# Model-based evaluation of alternative reactive class closure strategies against COVID-19

Quan-Hui Liu[1,12], Juanjuan Zhang[2,3,4,12], Cheng Peng[2], Maria Litvinova [5], Shudong Huang[1], Piero Poletti [6], Filippo Trentini [7], Giorgio Guzzetta [6], Valentina Marziano [6], Tao Zhou[8], Cecile Viboud [9], Ana I. Bento [5], Jiancheng Lv[1], Alessandro Vespignani[10,13], Stefano Merler [6,13], Hongjie Yu [2,3,4,13✉] & Marco Ajelli [11,13✉]

There are contrasting results concerning the effect of reactive school closure on SARS-CoV-2 transmission. To shed light on this controversy, we developed a data-driven computational model of SARS-CoV-2 transmission. We found that by reactively closing classes based on syndromic surveillance, SARS-CoV-2 infections are reduced by no more than 17.3% (95%CI: 8.0–26.8%), due to the low probability of timely identification of infections in the young population. We thus investigated an alternative triggering mechanism based on repeated screening of students using antigen tests. Depending on the contribution of schools to transmission, this strategy can greatly reduce COVID-19 burden even when school contribution to transmission and immunity in the population is low. Moving forward, the adoption of antigen-based screenings in schools could be instrumental to limit COVID-19 burden while vaccines continue to be rolled out.

[1] College of Computer Science, Sichuan University, Chengdu, China. [2] School of Public Health, Fudan University, Key Laboratory of Public Health Safety, Ministry of Education, Shanghai, China. [3] Shanghai Institute of Infectious Disease and Biosecurity, Fudan University, Shanghai, China. [4] Department of Infectious Diseases, Huashan Hospital, Fudan University, Shanghai, China. [5] Department of Epidemiology and Biostatistics, Indiana University School of Public Health, Bloomington, IN, USA. [6] Center for Health Emergencies, Bruno Kessler Foundation, Trento, Italy. [7] Dondena Centre for Research on Social Dynamics and Public Policy, Bocconi University, Milan, Italy. [8] Big Data Research Center, University of Electronic Science and Technology of China, Chengdu, China. [9] Division of International Epidemiology and Population Studies, Fogarty International Center, National Institutes of Health, Bethesda, MD, USA. [10] Laboratory for the Modeling of Biological and Socio-technical Systems, Northeastern University, Boston, MA, USA. [11] Laboratory for Computational Epidemiology and Public Health, Department of Epidemiology and Biostatistics, Indiana University School of Public Health, Bloomington, IN, USA. [12] These authors contributed equally: Quan-Hui Liu, Juanjuan Zhang. [13] These authors jointly supervised this work: Alessandro Vespignani, Stefano Merler, Hongjie Yu, Marco Ajelli. ✉email: yhj@fudan.edu.cn; marco.ajelli@gmail.com

The novel coronavirus disease 2019 (COVID-19) pandemic caused by severe acute respiratory syndrome coronavirus 2 (SARS-CoV-2) has dramatically changed the life of nearly every human on the planet. In Europe, Italy was the first country to experience the pandemic and it has been considered a natural experiment for large scale non-pharmaceuticals interventions. The first locally transmitted COVID-19 case in Italy was identified on February 21, 2020; since then, the country went through two distinct major epidemic waves. During the first wave, a national lockdown was put in place on March 11, 2020[1]. After the lifting of the lockdown on May 18, 2020[2], the number of COVID-19 cases remained relatively low throughout the rest of the spring and summer of 2020. However, after school reopening and further relaxation of control measures, a second major epidemic wave started in mid-September 2020. At that point, case isolation, contact tracing, and other social distancing measures (e.g., limited size of gatherings, closure of theaters and cinemas[3]) were still in place along with a newly established reactive class closure protocol based on active surveillance of students[4]. To counter the rapid rise of cases, a set of nationwide restrictions were imposed by the Italian government on October 14, 2020[5]. New restrictions included an extended mandatory use of face masks, reduction of opening hours or full closure of commercial/recreational venues, and partial or full suspension of in-person education. Control measures gradually increased in the following 3 weeks[6–8]. Since November 6, 2020, more restrictive measures were applied on a regional basis to further mitigate COVID-19 burden of the second wave[9], fueled in early 2021 by the more transmissible Alpha variant[10].

Italy is not an isolated example. A similar upsurge of COVID-19 cases right after school reopening in September–October 2020 was observed in several other European countries such as Finland, Ireland, Latvia, Belgium, and Slovakia[11,12]. Moreover, whether associated with school transmission or not, a decrease in the average age of cases was observed, with a larger fraction of cases in the school-age population[13,14]. A similar issue has been experienced in the US in early March 2021 as schools resumed in-person learning amidst rising incidence of the Alpha variant (e.g., Michigan[15]). While the virus is still circulating in 2021 in the form of a more transmissible variant (Delta)[16–18], understanding whether and how in-person education can be maintained is paramount.

The aim of this paper is twofold. First, we developed a computational model of SARS-CoV-2 transmission to estimate the contribution of the reactive school closure strategies implemented in Italy when historical lineages were circulating in the fall of 2020 to mitigate the second major COVID-19 wave and understand the reason of their limited effect. Second, we tested an alternative policy based on rapid antigen-based screening of students. We found that this strategy may have a considerably larger mitigation effect on SARS-CoV-2 spread, which may be crucial while COVID-19 vaccines continue to be rolled out throughout 2021 and beyond.

## Results

**Modeling SARS-CoV-2 transmission.** Based on detailed sociodemographic data, we developed a synthetic population of agents representative of the Italian population, whereby each agent in the model corresponds to an individual of the actual population[19]. The transmission of SARS-CoV-2 is modeled through the simulation of contacts between agents of the synthetic population in three social settings: (i) household members, (ii) schoolmates and classmates, and (iii) other individuals in the community (which includes both contacts between work colleagues in the workplaces and random encounters in the community). Details are reported in "Methods" and Supplementary Methods. The model allows the explicit simulation of the testing,

isolation, and quarantine strategies along with the reactive class-closure strategy, as implemented in Italy in the fall of 2020 when historical SARS-CoV-2 lineages were in circulation. Specifically, by drawing inspiration from the Italian case, we modeled the following interventions: (i) active syndromic surveillance: a symptomatic (non-student) individual has a probability (45% in the baseline analysis) of being tested through PCR; if positive (i.e., based on a draw from a Bernoulli distribution reflecting the sensitivity of PCR test), the individual is isolated at home for 14 days. Sick isolated individuals can transmit the infection to their household members only; and the members of households are quarantined at home for 14 days as well (95% probability to account for possible sub-optimal adherence); (ii) enhanced syndromic surveillance in schools: a symptomatic student isolates in the same way as scenario (i), but with a higher probability of being tested through PCR (95% in the baseline analysis); (iii) if a student is confirmed positive through a PCR test (either by the school symptomatic surveillance system or as a household member of an identified case), the student's class is closed for 14 days while teaching activities are maintained in the other classes of the same school (see Supplementary Methods for details and model parameters).

The model leverages data on COVID-19 natural history and SARS-CoV-2 transmission patterns observed in Italy. In particular, we calibrate the model to have a reproduction number (i.e., the mean number of secondary infections caused by a primary infector) of 1.1 when all schools were closed during the summer of 2020[20] and a household secondary attack rate of 51.5%[21]. Then, after schools reopened in mid-September 2020, the reproduction number $R$ increased to 1.3–1.9, depending on the Italian region; for instance, over the period October 2–8, 2020 $R$ was estimated to be ~1.3 in Sicily, ~1.5 in Lazio, ~1.7 in Veneto (also close to the national average), and ~1.9 in Lombardy[20]. This is similar to the increase estimated for the UK (namely an absolute increase of 0.2–0.7[22]). To simulate a situation close to that of September/October 2020 in Italy, we initialized the simulation with 5% of the population being immune to SARS-CoV-2[23]. As direct quantitative estimates of the contribution of school (or school-related) activities to the increase in the overall transmissibility are unavailable, we consider three scenarios. In the first scenario (F100), we kept the transmission rates in the household and community as estimated for the summer period and set the transmission in the school to obtain the target value of the reproduction number, which corresponds to attributing 100% of the observed increase of the reproduction number in September/October 2020 to school transmission. Overall, the total number of infections linked to school transmission in this scenario is 14.4–35.4% (while 41.2–51.4% are linked to household transmission and 23.4–34.2% to community; Supplementary Fig. 2). The second, more conservative, scenario assumes that 50% of the infections attributed to school transmission in F100 are in fact derived from school and the remainder 50% are due to increased transmission in the community, as other activities involving a predominantly young population had resumed[3]. In this scenario, the total infections linked to school is estimated to be 7.3–18.3% (Supplementary Fig. 2). Finally, the third scenario (F25) assumes that 25% of the transmission increase was linked to school transmission with the rest occurring in the community (4.0–8.7% of total infections are linked to school; Supplementary Fig. 2). Details about the model calibration are reported in "Methods" and Supplementary Methods; model parameters are reported in Supplementary Tables 1 and 2. All other nonpharmaceutical interventions adopted in Italy to limit transmission in the community or within schools are implicit as concerted strategies that result in the different values of $R$ (1.3, 1.5, 1.7, and 1.9)—overall transmissibility—and scenarios (F25, F50, and F100)—school relative contribution to transmissibility—explored in this study.

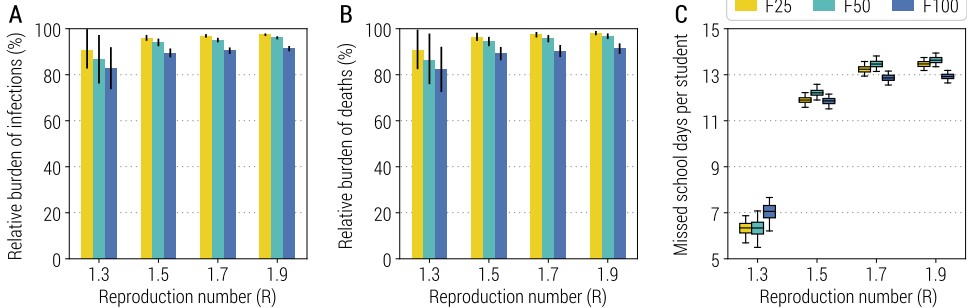

**Fig. 1 Impact of the reactive class-closure policy based on syndromic surveillance. A** Relative burden of the cumulative number of infections after 1 year as a function of the reproduction number and for different scenarios about school transmission contribution. The bar corresponds to the mean value, while the vertical line represents 95% quantile intervals; colors refer to the three scenarios F25, F50, F100. Parameters are as the baseline values reported in Supplementary Tables 1 and 2. Note that R is estimated in the absence of the class-closure strategy. The relative burden is defined as the estimated number of infections after 1 year since the introduction of the first infected individual with the class-closure strategy implemented, relative to the estimated number without the implementation of the class-closure strategy. For each scenario, 500 stochastic simulations were performed. **B** As in (**A**), but for the number of deaths. **C** Number of missed school days per student due to the reactive class-closure strategy. The boxplot indicates quantile 0.025, 0.25, 0.5, 0.75, and 0.975.

**Impact of the adopted reactive class-closure policy**. By forward simulating 1 year of epidemic, we estimate the infection attack rate (which includes all SARS-CoV-2 infections, independently of whether an individual develops symptoms or not) to decrease by less than 18% as compared to a counterfactual scenario with no surveillance in schools, regardless of the school transmission contribution scenario and reproduction number (Fig. 1A). Slightly lower reductions are estimated for the number of COVID-19-related deaths (Fig. 1B); other metrics related to the burden of COVID-19 are reported in Supplementary Methods (Supplementary Fig. 3). Despite this relatively small mitigation effect, the simulated reactive class-closure policy still entails a cost in terms of missed education: the range of estimates for the mean number of missed school days per student per year is between 6.3 days (95% CI: 5.7–6.9; CI indicates quantile interval in the whole manuscript) and 13.5 days (95% CI: 13.2–13.7) for F25, and between 7.0 days (95% CI: 6.2–7.7) and 12.9 days (95% CI: 12.6–13.2) for F100 (Fig. 1C). This would be comparable to a full school closure for ~7% of a school year (i.e., 200 school days). Importantly, this mitigation effect of the strategy is robust to the number of seeds used to initialize the epidemic (Supplementary Fig. 5).

The implemented reactive class-closure strategy is based on the premise of being able to timely identify cases either through symptomatic surveillance of students or through contact tracing. In the baseline scenario, we assumed that the probability of being tested is 95% for symptomatic students, and 45% for symptomatic non-student individuals, and the time intervals from symptom onset to sample collection and from sample collection to laboratory diagnosis are both 2 days (Supplementary Methods). To test the timeliness of this strategy, we looked at the number of SARS-CoV-2 infected individuals in an entire school rather than limiting to a class at the time when an infected student is identified, and reactive class closure is triggered. Compared with an average school size of 623.8 students and class size of 23.4 students[24], we estimated that the mean proportion of infectious students in each open class at the time when a class of the same school is reactively closed ranges from 1.0% (95% CI: 0.0–8.0%) to 7.7% (95% CI: 0.0–28.6%), for F50 and R = 1.3 and 1.9, respectively. This finding is very consistent across different school transmission contribution scenarios (Fig. 2A–C).

To explore whether and to what extent the adopted strategy could be improved or if its limited mitigation benefit is linked to its design, we analyzed alternative scenarios based on an improved testing capacity (in terms of probability of testing symptomatic students or symptomatic non-student individuals

and shorter time intervals from symptom onset to sample collection or laboratory diagnosis). The obtained results are consistent with those obtained for the baseline strategy (Fig. 3 and Supplementary Fig. 4), suggesting a structural weakness in the strategy design. These results are also confirmed when we assume a shorter mean incubation period, homogeneous susceptibility to infection by age, and when symptomatic individuals are assumed to be twice or four times more infectious than asymptomatic individuals (Supplementary Figs. 6–8). Moreover, the obtained results are also consistent in scenarios where: (1) the fraction of the initially immune population (either due to natural infection or vaccination) is increased to 20%, closer to the situation in Italy in early 2021[25], and (2) an age-dependent initial immunity is considered (Supplementary Figs. 9 and 10).

**Impact of a reactive school-closure policy**. The findings presented thus far suggest that, by the time that classes are reactively closed, outbreaks are already silently taking place in other classes of the same school due to the low probability of young individuals developing symptoms[21], thereby dramatically decreasing the mitigation effect of this strategy. This calls for the design and implementation of alternative strategies while vaccines are being rolled out.

Here we tested whether closing the entire school (as compared to the current policy entailing class-based closures) when a SARS-CoV-2 infected student is identified represents a better option. Our simulation results show that the mitigation effect of this alternative strategy is much larger than that of the 2020 policy, but at the cost of having schools closed for nearly the entire duration of the epidemic (Supplementary Fig. 11). This remains true when up to 20% of the population is initially immune (Supplementary Fig. 12). Moreover, this strategy fails to timely identify positive students thereby not addressing the main weakness of the baseline class-closure strategy.

**Impact of an antigen-based screening strategy**. The results presented so far call for the design of a reactive class/school-closure policy that goes beyond syndromic surveillance of students. To address this, we consider the potential introduction of screenings of the student population based on antigen tests. This type of testing has the advantages of allowing a quick turnaround time (minutes to hours) and lower costs relative to PCR tests[26]. It is worth noting that, in Italy, antigen rapid tests have been used alongside PCR tests to identify SARS-CoV-2 positive individuals since October 30, 2020[27].

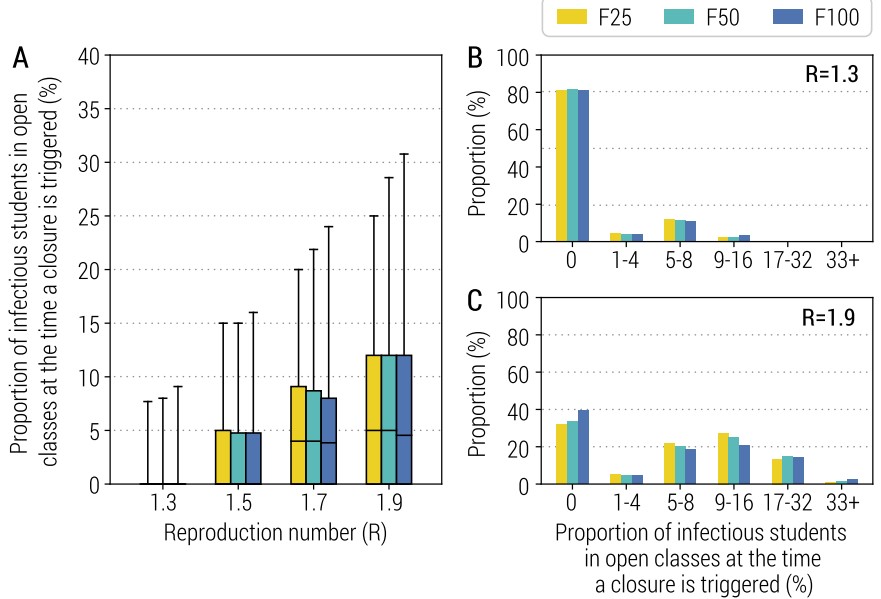

**Fig. 2 Infectious students at the time of class closure. A** Proportion of infectious students in all the open classes of the school (i.e., excluding the class triggering the class closure) for different values of $R$ at the time when a class closure is triggered. Parameters are as the baseline values reported in Supplementary Tables 1 and 2. Note that $R$ is estimated in the absence of the class-closure strategy. For each scenario, 500 stochastic simulations were performed. The boxplot indicates quantile 0.025, 0.25, 0.5, 0.75, and 0.975. **B** Distribution of the proportion of infectious students in all the open classes of the school (i.e., excluding the class triggering the class closure) for $R = 1.3$. **C** As in (**B**), but for $R = 1.9$.

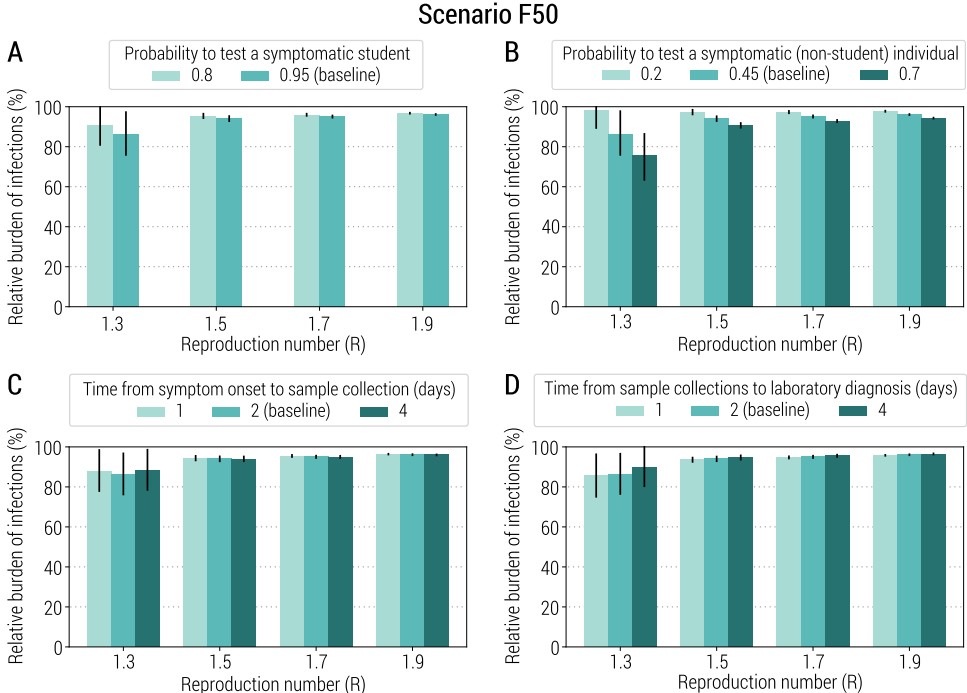

**Fig. 3 Sensitivity of the class-closure strategy relying on syndromic surveillance to changes in parameters regulating its implementation. A** Relative burden of the cumulative number of infections after 1 year as a function of the reproduction number and for different values of the probability to test a symptomatic student at school. The bar corresponds to the mean value, while the vertical line represents 95% quantile intervals. Parameters are as the baseline values reported in Supplementary Tables 1 and 2. Note that $R$ is estimated in the absence of the class-closure strategy and the scenario considered is F50. The relative burden is defined as the estimated number of infections after 1 year since the introduction of the first infected individual with the class-closure strategy implemented, relative to the estimated number without the implementation of the class-closure strategy. For each scenario, 500 stochastic simulations were performed. **B** As in (**A**), but for the probability to test a symptomatic (non-student) individual in the community. **C** As in (**A**), but for the time from symptom onset to sample collection. **D** As in (**A**), but for the time from sample collection to laboratory diagnosis.

We define an alternative strategy based on repeated screening of students (regardless of symptoms) with rapid antigen testing, while the symptomatic surveillance of the general population remains in place unaltered. We conducted a meta-analysis of the literature to obtain estimates of the sensitivity and specificity of antigen tests, which are 69% (95% CI: 41–97%) and 99% (95% CI: 97–100%), respectively (see Supplementary Methods for details). We tested three different screening schedules: antigen-based tests provided to all students every 3, 7, or 14 days. If a student is found to be SARS-CoV-2 positive either through symptomatic surveillance or antigen screening, the class of such a student is closed for 14 days while the other classes in the school remain open (see Supplementary Methods for details).

We estimate that the strategy based on a weekly antigen screening of students can reduce COVID-19 infection attack rate between 65 and 83% for scenario F100 and $R$ up to 1.9 (Fig. 4A). Other metrics of COVID-19 burden are reported in Supplementary Fig. 13. For scenarios F50 and F25, the impact of this strategy greatly depends on $R$, with reductions ranging from 20% to 70% (Fig. 4A). The number of missed school days per student greatly depends on $R$ as well (Fig. 4B). The minimum number of missed school days per student is estimated for $R = 1.3$ and F100 with 23.6 (95%CI: 18.1–29.3) days, while the largest value is estimated for $R = 1.7$ and F50 with 60.3 (95%CI: 56.3–64.6) days. It is worth noting that there are several forces at play that determines the number of missed days. While larger values of $R$ result in a larger number of infections (and thus the larger the number of triggered class closures), it also leads to quicker epidemics spreads, thus reducing the time for class closures to be triggered.

The observed greater mitigation effect of the antigen-based screening strategy as compared to the syndromic-based screening strategy is due to the much better capacity to identify infectious students in a timely manner. For Scenario F50 and a once-a-week testing schedule, the mean proportion of infectious students in the open classes at the time when a class of the same school is reactively closed never exceeds 0.2% (95%CI: 0.0–4.3%) students for $R \leq 1.5$, and 1.1% (95%CI: 0.0–10.0%) students for $R = 1.9$ (Fig. 4C). Moreover, almost no infectious students can be detected in other open classes when one class of the same school is closed (Supplementary Fig. 14). These results are also confirmed when we assume that the sensitivity of the antigen test depends on the time since infection and test administration (Supplementary Fig. 15).

By decreasing the testing frequency to once every 2 weeks, we estimate a decrease in the effectiveness of the strategy (Fig. 4D and Supplementary Fig. 16), while the number of missed school days per student never exceeds 43.6 days (Fig. 4E). This is likely due to the generation time being lower than 7 days[28–30], suggesting that the testing frequency should be comparable to or shorter than the generation time. We performed sensitivity analyses where we consider a lower coverage of the policy. We assumed 50%, 75%, or 90% of students being tested (rather than 100% considered in the baseline analysis). In this case, the strategy assuming 50% screening probability led to 38.2–80.4% higher burden, depending on the value of $R$, as compared to that assuming 100% screening probability (Fig. 4F). In Supplementary Methods, we report the results of sensitivity analyses on the daily number of identified positive individuals necessary to trigger the antigen screening in schools (Supplementary Fig. 17) and on alternative levels of pre-existing immunity (Supplementary Fig. 18).

## Discussion
Previous studies have investigated the impact of proactive school-closure strategies in reducing SARS-CoV-2 transmission[31–40]. We provide a quantitative assessment of reactive class closures

implemented in Italy in the fall of 2020, in combination with contact tracing and other social distancing measures, to provide a potential explanation of why the adopted strategy was not successful in preventing a second nationwide COVID-19 wave. We propose and evaluate the effectiveness of an alternative strategy that could be applied while vaccines are rolled out. Our modeling results suggest that using syndromic surveillance to trigger case isolation, contact tracing, and the reactive closure of classes with a confirmed COVID-19 infection has a limited impact in mitigating COVID-19 burden (reduction of ~10% as compared to no school interventions). In fact, Italy had to rely on a set of restrictions at the regional level and closure of all schools for specific ages as a response to the spread of the second COVID-19 wave caused by historical lineages in the fall of 2020[9].

Our results show that the reactive class-closure strategy implemented in the fall of 2020 has a limited potential in mitigating COVID-19 burden. This result is consistent when considering a wide set of sensitivity analyses on COVID-19 epidemiological characteristics (e.g., incubation period, age-specific susceptibility to infection, infectiousness of asymptomatic individuals relative to symptomatic ones, age-dependent heterogeneity in population immunity, incubation), parameters regulating the implementation of the strategy (probability of testing symptomatic students and symptomatic non-student individuals, time intervals from symptom onset to sample collection or laboratory diagnosis), and model parameterization (daily imported infections to initialize the epidemic). Nonetheless, it is possible that the relative infectiousness of asymptomatic individuals could play an important role in the school transmission since children are more likely to be asymptomatic[21]. However, our sensitivity analysis that considers asymptomatic individuals being two or four times less likely to transmit the virus[41], shows a similar mitigation impact of the analyzed class-closure strategies.

Our results show that the deployment of antigen tests to perform a routine screening of the student population has the potential to successfully mitigate SARS-CoV-2 spread not only in schools, but in the community at large. This hypothesized policy is estimated to greatly reduce COVID-19 burden $R$ up to 1.3–1.5, depending on the scenario. Multiple reasons contribute to the (estimated) success of this strategy. First, this strategy allows the identification of asymptomatic infections in students—a large fraction of infections in this segment of the population[21]. Second, the identification of asymptomatic students prevents them from transmitting both in the school environment and in the community. Third, the identification of infected students triggers prompt quarantining of their household member, potentially preventing new chains of infections outside the school setting. Finally, the rapid turnaround of antigen-based tests compared to PCR allow for a timely withdrawal of infected individuals from the transmission process. Our study supports the importance of student screening and testing for limiting silent transmission in schools and thus reduce the overall infection attack rate. As reported in other studies[42], the rapid identification of silent infections among children is a key tool for mitigating SARS-CoV-2 spread.

We note that, in our study, we do not address questions regarding the logistics, feasibility, and acceptability of this strategy. Indeed, an adequate stockpile of antigen-based tests is needed as well as the appropriate logistics surrounding capacity to collect samples from students (e.g., while at school), and compliance to the policy. Moreover, it remains to be determined whether this strategy is cost-effective, accounting for direct and indirect costs associated with the illness, PCR and antigen-based testing, loss of productivity, and education. Sporadic attempts have been conducted in specific Italian locations[43] and in several

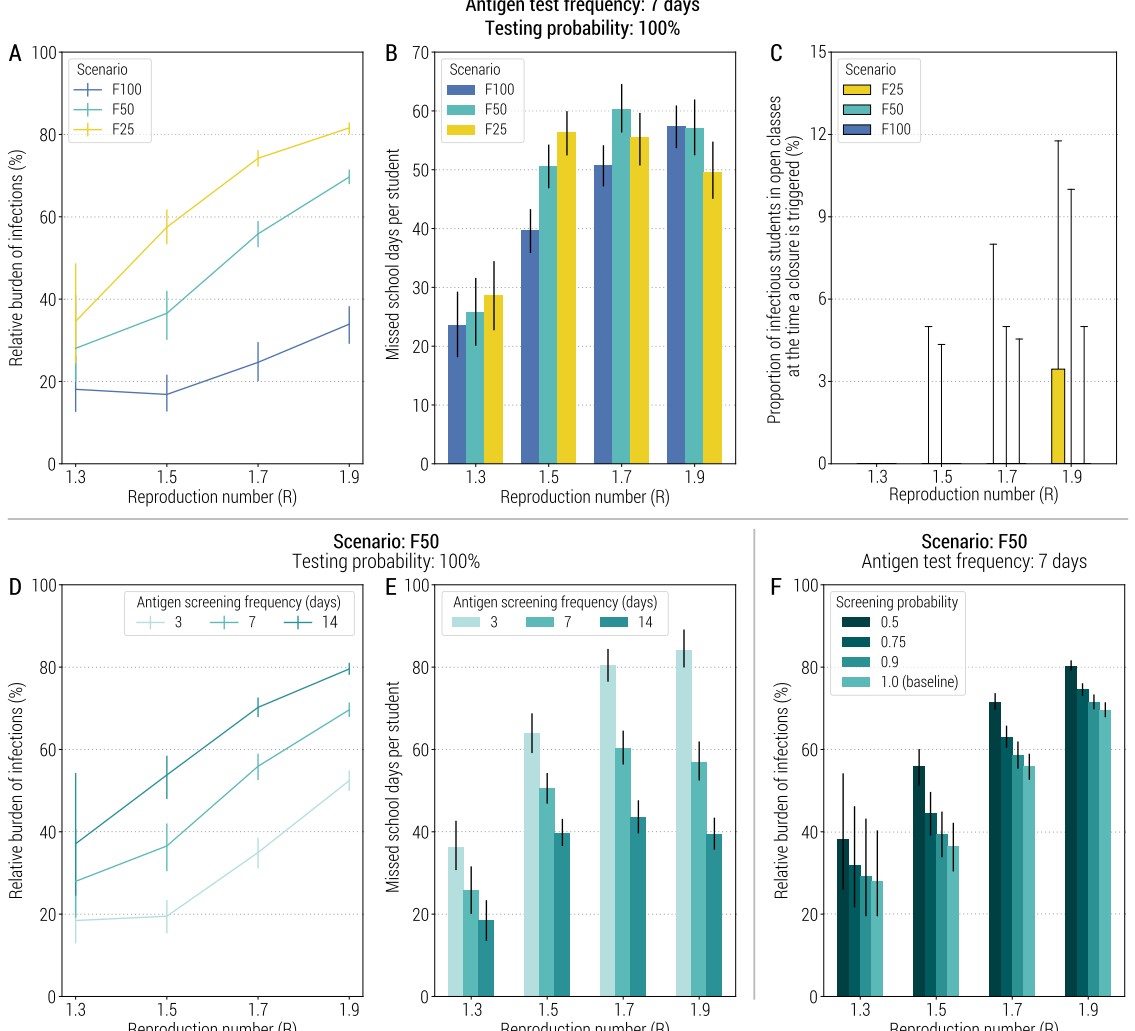

**Fig. 4 Impact of reactive class-closure policy relying on antigen screening. A** Relative burden of the cumulative number of infections after 1 year as a function of the reproduction number and for different scenarios about school transmission contribution. The line corresponds to the mean value, while the vertical line represents the 95% quantile intervals; colors refer to the three scenarios F25, F50, and F100. The fraction of immune population at the beginning of epidemic is set at 10%, the probability of testing a student at school with the antigen test is 100%, the frequency of the antigen testing is weekly; other parameters are as the baseline values reported in Supplementary Tables 1 and 2. Note that $R$ is estimated in the absence of the class-closure strategy. The relative burden is defined as the estimated number of infections after 1 year since the introduction of the first infected individual with the class-closure strategy implemented, relative to the estimated number without the implementation of the class-closure strategy. For each scenario, 500 stochastic simulations were performed. **B** Number of missed school days per student due to the reactive class-closure strategy. The bar corresponds to the mean value, while the vertical line represents 95% quantile intervals. **C** Proportion of infectious students in all the open classes of the school (i.e., excluding the class triggering the class closure) for different values of $R$ at the time when a class closure is triggered. The boxplot indicates quantile 0.025, 0.25, 0.5, 0.75, and 0.975. **D** As in (**A**), but for scenario F50 and by varying the frequency of testing (every 3, 7, or 14 days). **E** As in (**B**), but for scenario F50 and varying frequency of testing (every 3, 7 or 14 days). **F** As in (**A**), but for scenario F50 and by varying the probability of antigen testing (50%, 75%, 90%, and 100%). The bar corresponds to the mean value, while the vertical line represents 95% quantile intervals.

US universities[44,45]. Moreover, in Slovakia a population-wide rapid antigenic screening strategy proved to be feasible and highly effective[46]. Compared to the reactive class-closure strategy based on syndromic surveillance, antigen-based screening is more effective but entails greater costs in terms of missed school days. Moreover, more frequent screening results in higher number of missed school days. The main contribution to this trend comes from the fixed duration of quarantine (14 days) of all students of a closed class, including those who tested negative. Also, students identified as positive during the screening had to remain in isolation for 14 days, regardless of whether they are still infectious or not. Repeated PCR testing of isolated and quarantined students would allow them to get back to school once they are confirmed

as SARS-CoV-2 negative, decreasing unnecessary prolonged isolations and quarantines[47]. In this case, a strategy based on frequent screening may allow to both curtailing transmission and its associated burden in terms of missed school days. The number of missed school days could be further curtailed by requiring PCR confirmation of antigen positive samples as trigger for class closures.

We cannot rule out the possibility that, together with schools reopening and an increase in work and community activities[48], climatic factors may have contributed to the increase of SARS-CoV-2 transmission observed in the fall of 2020[49]. Regardless, the relative contribution of school transmission to the observed trend remains elusive. As such, in our modeling work,

we have considered three alternative scenarios on this key parameter. Although the main message of the study remains unchanged, the scenarios highlight noteworthy quantitative differences in the mitigation effect of the proposed policies, calling for further research on this subject.

We consider only random mixing in the school and within each class, where the relative weight of these two components is derived from a pre-pandemic contact-survey study[50]. However, we acknowledge that the school structure and organization (e.g., phased school entries, limited group size in public areas of the school) has been deeply changed due to the COVID-19 pandemic. Unfortunately, as of October 2021, we are not aware of any study on the within-school and within-class contact network of students in the COVID-19 era in Italy. Moreover, the mixing patterns in the community have remarkably changed as well. For example, mass gatherings (e.g., attending sport events, disco, cinema) were either banned for most of the duration of the pandemic or the capacity has been reduced, restrictions were imposed about the maximum number of people sitting at the same table in restaurants or allowed to enter commercial buildings at the same time, etc. As such, we kept the model as simple as possible, assuming homogenous mixing both in the school and community, although this represents a "first-order" approximation of the much more complex network of interconnections (e.g., between students attending different schools)[51]. Nevertheless, the model has the flexibility to incorporate a more realistic representation of mixing patterns during the pandemic should new data become available for the focus location. Mixing patterns as well as social, behavioral, and cultural characteristics of the population (e.g., number of persons per room in the house, which household member serves as a primary caregiver inside the household) have the potential to shape the household secondary attack rate. As such, we relied on an estimate available for Italy (about 50%[21]) rather than estimates derived from contact tracing studies conducted in other countries[52].

We built a synthetic population of about 0.5 million individuals (roughly the size of the sixth largest Italian city), rather than simulating the whole Italian population. The main reason that prevents us from performing a country-scale analysis is the spatial connectivity of the Italian population during that phase of the pandemic. In the fall of 2020, several restrictions to the mobility between regions and between provinces were implemented with different levels of intensity over different time windows; moreover, the behavior of the population in terms of travel patterns was farm from pre-pandemic level. Overall, we believe that we do not have enough data to provide a robust representation of the Italian mobility to develop a detailed national-scale Italian model. The second factor that advises us against using a country-level model is the computational time needed to run all the analyses. Thus, we built a synthetic population well representative of a large Italian city.

The developed model is based on a synthetic population of social interactions of (a subsample of) the Italian population[53] and on Italy-specific data on COVID-19 epidemiology (infection fatality ratio[54], population immunity[23], hospitalization rates[55], etc.). Nevertheless, the introduced modeling framework is flexible, able to be tailored to other countries to provide insights on the design of COVID-19 control strategies. We do not explicitly model every single measure adopted in Italy to limit transmission in the community (e.g., ban of mass gatherings, closure of cinemas, use of masks) or within school (e.g., desk distancing, mandatory use of masks). These measures are implicit as concerted strategies that result in the different values of $R$ (1.3, 1.5, 1.7, and 1.9)—overall transmissibility—and scenarios (F25, F50, and F100)—school relative contribution to transmissibility—explored

in this study. However, our model could be extended to explicitly simulate additional single interventions (such as workplace closure, partial lockdowns, curfews) that can be adopted in conjunction with the proposed strategy, as well as to simulate the parallel rollout of COVID-19 vaccines and changing of its major target groups. Moreover, our model is flexible to be extended to the screening in other contexts (e.g., workplaces) to provide insights on alternative mitigation/containment strategies outside of school context.

Moving forward, should logistic challenges be overcome, and antigen screening of students gain traction among the public, it could speed up the "return to normal", while vaccines continue to be rolled out with booster doses and potentially vaccines tailored to emerging variants. The adoption of this policy could be a game-changing approach, especially as the emergence of new SARS-CoV-2 variants has illustrated the necessity to prepare for the prolonged co-existence with the virus circulation. While the cross-protection from vaccination and natural immunity will hopefully bring the effective reproduction number closer to one (e.g., close to influenza levels), it will be key to identify supplementary measures to guarantee safe in-person education in the long run.

## Methods

**Synthetic population**. We built a synthetic population of about 0.5 million individuals matching the sociodemographic structure of the Italian population. Every individual of the synthetic population has an associated age, belongs to a certain household, and attends a certain school (and a certain class within the school) if it is a student. School attendance, school sizes, class sizes, and ratio of teacher-to-student are derived from actual data. Details on the construction of the synthetic population are reported in Fumanelli et al.[53]. The synthetic population enables us to characterize four different social settings where contacts can occur: home, school, class within school, and the community (which includes any other contact). A schematic representation is shown in Fig. 5A. Given the lack of data about the relative risk of a workplace contact with respect to a community contact, we did not distinguish them in the model and thus a specific layer for workplace contact only was not included. Homogeneous mixing is assumed among individuals in each instance (i.e., a specific household, school, and class) of each social setting. Details are reported in Supplementary Methods.

**Model of SARS-CoV-2 transmission and COVID-19 burden**. We developed an individual-based mechanistic model of SARS-CoV-2 transmission for different social settings to estimate the impact of reactive class-closure policies in mitigating COVID-19 burden. Briefly, SARS-CoV-2 transmission is simulated according to an SLIR (susceptible, latent, infectious, removed) scheme, where infectious individuals are further divided into pre-symptomatic, symptomatic, and asymptomatic individuals (Fig. 5B). If a susceptible individual $i$ is connected with an infectious (either symptomatic, pre-symptomatic, or asymptomatic) individual $j$, the susceptible individual can acquire the infection with a setting-specific probability. The settings considered in the model are household, class, school, and community (which accounts for any contact not occurring in the aforementioned settings). The model also accounts for age-specific susceptibility to SARS-CoV-2 infection (as estimated in a meta-analysis by Viner et al.[52]). The incubation period follows a Gamma distribution with a mean of 6.3 days and a standard deviation of 4.3 (shape = 2.08, rate = 0.33)[30]. We considered transmission to start 2 days before symptom onset[30,56]. The duration of the infectious period was chosen such that distribution of the serial interval in the simulations match the distribution estimated for Italy (mean 6.6 days)[57,58]. The probability that an infected individual develops respiratory symptoms and/or fever follows the age-specific ratios estimated for the Italian population[21]. Specifically, it is defined as symptomatic if they had upper or lower respiratory tract symptoms or fever of 37.5 °C or higher, and respiratory symptoms included dry cough, dyspnea, tachypnea, difficulty breathing, shortness of breath, sore throat, and chest pain or pressure[21]. Details about the model and its calibration are reported in Supplementary Methods. A schematic representation of the transmission model is shown in Fig. 5.

To estimate COVID-19 burden, we leveraged the estimation of the infection hospitalization ratio, infection critical disease ratio, and infection fatality ratio obtained for Italy[21,54,55] (Supplementary Table 2). We applied them to the number of daily new infected individuals provided by the transmission model. Symptomatic individuals are instead calculated as described above.

**Simulated interventions**. The model explicitly simulates case isolation (in the place of residence), quarantine of household contacts (in the place of residence), and a reactive class-closure policy based on that implemented in Italy in the

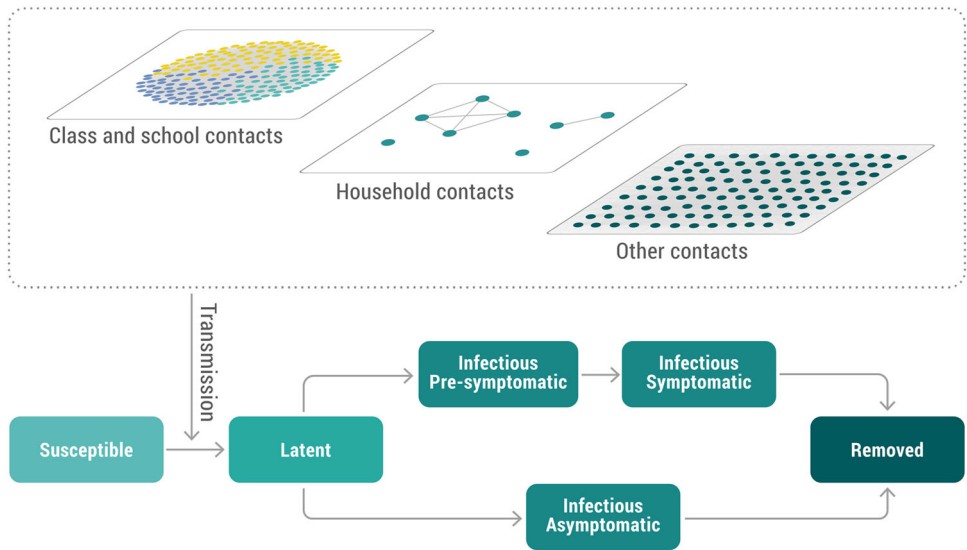

**Fig. 5 Transmission model and network structure.** Schematic representation of the transmission model and of the contact network.

2020–2021 school year. Case isolation and contacts quarantine are triggered by symptomatic individuals. Symptomatic students are tested with a 95% probability. Testing of symptomatic students was mandatory, but we considered that 5% of student population may not comply. For the non-student population, symptomatic testing was not mandatory in Italy (except for specific workplaces such as health care and nursing home workers). As such, we considered that a symptomatic non-student individual has 45% probability of being tested. This parameter is set so that the case ascertainment ratio for any symptomatic individual in the overall population results to be 31%, matching the value reported in Marziano et al.[23]. The sensitivity of the RT-PCR test depends on the delay between the date of infection and test according to the estimates provided in Kucirka et al.[59]. After sample collection and before the test result is obtained (2 days on average[60]), symptomatic individuals are precautionary quarantined in their place of residence and, if infectious, they can transmit to their household contacts only. Regardless of whether the positive individual is a student or not, they are isolated at home for 14 days starting with the date of laboratory confirmation. The household members of a positive individual are tested with RT-PCR and are quarantined at home for 2 weeks starting from the date of laboratory confirmation. Although mandatory, we considered a 95% compliance rate.

Regarding reactive class closures, in the main analysis resembling the policy applied in Italy, they are triggered by the identification of a positive symptomatic student. If that is the case, teaching activities for the class of the positive student are suspended for 14 days, while the other classes of the school remain open. We also considered an alternative (hypothetical) scenario where the same class-closure policy is triggered by the identification of a positive student through rapid antigen test screening. In this scenario, we consider that a fraction of students ranging from 50 to 100% is tested either every 3, 7, or 14 days.

**Model calibration and initialization.** We calibrated the model based on the epidemiological evidence on COVID-19 in Italy from the summer of 2020 until the fall of the same year. In particular, over the summer of 2020, schools were closed for the regular summer break and the net reproduction number ($R$) was estimated to fluctuate around 1.1[20]. In addition, a study based on Italian contact tracing data has estimated a household secondary attack rate of 51.5%[21]. As such, we simulated epidemics assuming no transmission in school and estimated the transmission rate in household and in the community (representing all contacts other than those in the household and at school) to obtain the reproduction number and household secondary attack rate reported above. All other model parameters (e.g., probability of developing symptoms by age, length of the incubation period) were taken from the literature (see Supplementary Table 1). Note that in this first step of the model calibration procedure we have two free (independent) parameters to estimate and two (independent) epidemiological indicators to match. Simulations are initialized considering that 5% of the population was already infected during the first wave[23] and the daily number of newly infectious individuals imported from outside the study area are sampled from a Poisson distribution of mean 1.34, based on the estimates reported by the national surveillance system (i.e., 0.027 imported cases per day per 10,000 individuals)[60].

In mid-September 2020, after school reopened after the summer break, the reproduction number sharply increased to 1.3–1.9 in the different Italian regions[20]. We do not have a direct quantitative estimate of the contribution of school (or school-related) activities to this increase in the overall transmissibility. As such, we considered three scenarios assuming a different contribution of schools (namely, F100, F50, and F25). This second step of the calibration procedure consisted in

determining the transmission rate in school to obtain different transmission scenarios. First, we defined a scenario where the entire increase of $R$ is ascribed to school transmission (scenario F100). In this case, we calibrated the transmission rate in school to reproduce four different values of the reproduction number (1.3, 1.5, 1.7, and 1.9), while keeping the transmission rates in household and community to the values derived in the first step. For each value of $R$, we thus have one configuration of parameters leading to the desired estimate. Note that the relative transmissibility within a class and at school corresponds to the ratio between contacts with classmates and contacts with schoolmates (excluding classmates) estimated in a pre-pandemic contact survey[50]. To calibrate scenarios F50 and F25, we adopted the following procedure. First, we recorded the number of infections generated in schools for scenario F100 and each value of $R$; let us define this number of infections as $f$. In scenarios F50 and F25, we considered that only a certain fraction of the transmission increase observed in Italy after the reopening of schools was linked to school transmission. In particular, in scenario F50, we consider that 50% of $f$ was related to school transmission (and 25% of $f$ in scenario F25). As such, while we kept fixed the household transmission rate, we searched for the transmission in the community and in school to match the desired value of $R$ and the number of infections linked to school transmission. Overall, also in this step there is a single configuration of the two model parameters (school transmission rate and community transmission rate) that matches the two conditions (reproduction number and fraction of infections at school as compared to scenario F100).

**Data analysis.** For each scenario, 500 stochastic simulations were performed. The number of model simulations used in the analysis was empirically determined to guarantee the stability of the results. We defined 95% credible intervals as quantiles 0.025 and 0.975 of the estimated distributions. Model simulations were performed in C and model output was analyzed in Python (version 3.9.7).

**Reporting summary.** Further information on research design is available in the Nature Research Reporting Summary linked to this article.

## Data availability

The data needed to reproduce this study is available on Zenodo[61].

## Code availability

The code used in this study is available on Zenodo[61].

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

## Acknowledgements

The authors would like to thank Nicole Samay for her assistance in preparing the figures. Q.-H.L. acknowledges funding from the National Natural Science Foundation of China (No. 62003230), Chengdu Science and Technology Bureau (No. 2020-YF05-00073-SN), the Fundamental Research Funds for the Central Universities (No. 1082204112289), the Science and Technology Department of Sichuan Province (No. 2020YFS0009). Q.-H.L., S.H., and J.L. acknowledge funding from the 111 Project under grant agreement B21044. T.Z. acknowledges funding from the National Natural Science Foundation of China (No. 11975071, No. 61673085). J.L. acknowledges funding from the National Science Fund for Distinguished Young Scholars (No. 61625204). P.P., F.T., G.G., V.M., and S.M. acknowledge funding from the European Union Grant 874850 MOOD (cataloged as MOOD 000). H.Y. acknowledges funding from the Key Program of the National Natural Science Foundation of China (No. 82130093). M.L., A.V., and M.A. acknowledge funding from the Cooperative Agreement number NU38OT000297 from the Centers for Disease Control and Prevention (CDC) and the Council of State and Territorial Epidemiologists (CSTE). The study does not necessarily represent the views of CDC and CSTE. The funders had no role in the design and conduct of the study; collection, management, analysis, and interpretation of the data; preparation, review, or approval of the manuscript; and decision to submit the manuscript for publication.

## Author contributions

A.V., S.M., H.Y., and M.A. designed the research; Q.-H.L. and M.L. developed the model; Q.-H.L. performed the simulations; Q.-H.L., J.Z., and C.P. analyzed the data; Q.-H.L., J.Z., C.P., M.L., S.H., P.P., F.T., G.G., V.M., T.Z., C.V., A.I.B., J.L., A.V., S.M., H.Y., and M.A. interpreted the results; J.Z., M.L., and M.A. wrote the manuscript; P.P., G.G., C.V., A.I.B., A.V., and H.Y. edited the manuscript.

## Competing interests

A.V. reports grants from Metabiota Inc., outside the submitted work. H.Y. has received research funding from Sanofi Pasteur, GlaxoSmithKline, Yichang HEC Changjiang Pharmaceutical Company, and Shanghai Roche Pharmaceutical Company. M.A. has received research funding from Seqirus. None of those research funding is related to COVID-19. All other authors report no competing interests.
