## [Peer Review File · Nature Communications]

Model-based evaluation of alternative reactive class closure strategies against COVID-19REVIEWER COMMENTS

Reviewer #1 (Remarks to the Author):

This is a nice analysis that uses an individual based model to estimate the impact of various school-oriented mitigation strategies on the overall transmission of SARS-CoV-2 in the community. They use a previously developed synthetic population to model transmission with explicit account of households and schools. The model structure is generated such that it resembles related data, but does not represent any particular population explicitly. They find that reactive class closure and interventions based on regular screening is more effective than quarantining based on syndromic testing. This paper addresses an important question; the approach is appropriate and the study is clearly communicated and well thought through. I have some comments that I recommend the authors address before this article is published.

Major comments:

1. Code availability:

It is becoming more usual that code is published alongside academic work for full transparency. I would appreciate access to the code to allow a full review of the work.

2. Discussion of limitations.

There is currently very little discussion of how the model choices affect the results. Since this is an IBM, the outcome is very sensitive to the model specification and parameterisation. I would expect to see a detailed justification (including some in the main text) of these model choices and discussion of their impact on results. (I refer to some of these in more detail below)

Epidemiological parameterisation:

The model relies heavily on its specification and parameterisation. I think the authors could be clearer about the underlying epidemiological model in the main text. I also have some concerns about some of the epidemiological parameter ranges chosen. I would like to see more discussion of these choices and how they might affect the results.

3. Given the recent changes in epidemiology due to new variants (particularly B.1.1.7 and B.1.617.2), I think it is now essential to discuss which variant this paper intends to study – although I imagine this was not the case when the work was carried out. A frustrating factor in modelling during a fast-moving pandemic.

4. One key parameter, which is likely to affect results, is susceptibility of school aged children relative to adults. Broadly, this is still not clear in literature and therefore must be discussed as part of a work looking at interventions targeted at children. I would have expected this issue to be raised when presenting parameterisation in the main text, as well as discussion of how this might impact the results. The authors have based their estimate on the results of one study that uses data from very early in the pandemic. There have been a large number of studies using more recent data which may impact this choice and in turn the outcome of the study. Ideally, I would like to see more estimates applied in a sensitivity analysis, I recommend a discussion of this issue and how results might be affected.

5. Household SAR – it seems that this was derived from work unrelated to household attack rate. Household SAR has been estimated in a number of studies with results quite different from this (but again this depends on the variant the authors wish to model). Russel Viner has a systematic review of estimates. I recommend that this choice is reviewed, although I am not clear on the implications for the results.

Minor comments:

1. The authors present the number of students infected before closing a class but it might be interesting to report the proportion of these in other classes.

2. It is interesting that more effective interventions (more frequent screening) results in more

missed days, do the authors have a sense for when the interventions become effective enough that missed days reduce.

3. The authors use a figure of 50% test rate for symptomatic cases in the community – is there a source for this?

4. Have the authors considered using within-school contact data from previous contact surveys to parameterise the model?

5. The model assumes homogenous mixing in the non-school, non-household population. This might impact the results as it may be expected that contacts cluster spatially which likely correlates with school populations. Could the authors comment on this? You might find this helpful for discussion of how schools interact through households <https://doi.org/10.1038/s41467-021-22213-0>.

6. Line 95 - presumably the student isolates in the same way as scenario i. This could be clearer.

7. line 130: This seems more like a discussion point. Could be briefer here and more detailed in discussion. – there are a number of similar paragraphs in the results section.

8. I find the plots presenting the relative change in infections a bit un-intuitive – at first glance it looks like they are more effective for higher R values. Perhaps invert the axis?

9. I cannot find where you state the number of iterations of the model. I think this needs to be stated in each Figure caption to reflect the filtered samples.

10. Excluding simulations with under 5% infected to filter out spontaneous extinction seems sub-optimal (although I can see why you would do it). Could you instead simulate infections imported into the otherwise closed population (as randomly occurring infections) to protect against extinction altogether? This seems closer to reality.

Reviewer #2 (Remarks to the Author):

An age stratified agent-based model was used to quantify the effects of testing through RT-PCR and school closures on community level transmission, hospitalization, and death. The results of the model suggest that screening students frequently with agent-based tests can be an effective measure to mitigate the burden of COVID-19. In general, I did not find too many major issues. As I read the manuscript, I felt there were parts of the methods and results that could be clarified for readers.

Major

1. The sensitivity of the RT-PCR and antigen test was assumed to be static over the different stages of infection. Evidence suggests that the probability of detection is dependent on the stage of disease [Hellewell et al (2021) Estimating the effectiveness of routine asymptomatic PCR testing at different frequencies for the detection of SARS-CoV-2 infections, BMC Medicine ; Kucirka et al (2020) Variation in false-negative rate of reverse transcriptase polymerase chain reaction–based SARS-CoV-2 tests by time since exposure, Annals of internal Medicine]. With the fixed sensitivity of test, the effectiveness of testing is possibly over-estimated.

2. The specificity of the antigen test was estimated to be 99%, and then later assumed to be 100% for simplicity of the model. The issue here is that false positives will have implications on the specified interventions. Given the sheer volume of tests that would be conducted under the screening process, this assumption of 100% will have implications on the results. For example, conducting 100 tests on non-infected individuals that has a specificity of 99.9%, there is a 0.095 probability of at least one false positive.

3. The incubation period was fixed at five days. Given some interventions are triggered by the presence of symptoms, accounting for heterogeneity in the incubation period could have a large impact on the level of uncertainty of the results. One could conduct a sensitivity analysis on the duration of the incubation period, integrate individual heterogeneity of the duration of the incubation period with a statistical distribution, or both (i.e. distributions with different means).

Minor

1. Line 86 – 87 is repetitive of prior lines 83-86.
2. Line 107: The mention of a 0.2-0.7 increase should be specified as an absolute increase to limit any confusion if it is relative.
3. Instead of testing every student at one time one per week, the scenario of testing 1/7 of the student population every day was considered. This assumption implies that the school calendar was not considered. Given the flexibility of an agent-based model, this minor issue can easily be addressed. I expect that this small formality will not have substantial impact on the results.
4. Line 404: The model is said to be simulated according to an SIR model, but the SARS CoV-2 is better defined by the SEIR model structure.
5. Supplement pg 4: Infection is stratified into the infectious pre-symptomatic period, infectious symptomatic, infectious asymptomatic and recovered. However, there is no indication of the latent period (where the individual is infected but not infectious). It is unclear if this low level of infectivity is integrated into the parameter ϕ (infectiousness of individual j at time t), as I could not locate this function in Table S1. It is later specified to be chosen to reflect the generation time. This issue might be easily cleared up with an addition of a Figure to the SI.
6. Figure 2. The aspects of the box plot are not described in the caption. I noticed it was specified in Figure 1 that it carries out, but this statement should be repeated for each caption.
7. The application of some of the probabilities should be clarified. For example, there could be confusion when sampling whether someone develops clinical symptoms or not and then later determining hospitalization and ICU (e.g. if my Bernoulli trial implies no symptoms, then does not seem reasonable that I go to the hospital). My impression is that these probabilities are used to assess hospitalization/deaths at the end of the simulation based on the cumulative number of infections. These calculations should be clarified in the SI.
8. Simulations were initialized with a single case, but it was unclear whether simulations that quickly went extinct were discarded or not; and if not, the frequency in which early extinction occurred.
9. In the text, it is unclear the number of ensembles the statistics are based on.
10. Figure 4B/E: I am particularly interested in the trend in the missed school days per student for the different scenarios as the effective reproductive number increases. For example, when $R=1.5$ the F50 scenario has fewer missed school days than the F25 but when $R=1.9$ the F50 scenario has more missed school days than the F25, but the F100 scenario is always substantially lower. More interestingly is the trend with antigen testing where the different testing frequencies have more missed school days under the different R values. I think it would be helpful to readers to briefly discuss why this trend is occurring.
11. For the sensitivity analysis regarding immunity, it was assumed that 20% immunity was attained in all age classes. If immunity was shifted more so to the adults but still obtaining 20% immunity (i.e. if we think the children were sheltered from infection due to school closures), I expect transmission will become more prominent within the school

Reviewer #3 (Remarks to the Author):

In this manuscript, the authors use an individual-based model to evaluate the impact of school closure on COVID-19 disease transmission. There have been a number of inconsistent studies regarding the efficacy of school closure, with inconsistencies arising from model assumptions and structures. In this manuscript, the described model is incredibly elaborate, and takes into account

significant population heterogeneities that affect disease transmission and model outcomes. I commend them on their model structure and formulation, and have confidence in their results.

The authors find that school closures do reduce overall disease burden, but only to a mild degree (<15%), but can have a substantial impact on missed school days. In my interpretation, this is a very important point as missed school days may have significant socio-economical impact on the student as well as the general community, where this impact is possibly to be realized over the next few months or years.

Overall, I recommend this manuscript for publication but have a few minor concerns.

- The authors assume the infectiousness of asymptomatic individuals/students to be 100% relative to symptomatic infection. I am not sure if this assumption is justified (see: <https://doi.org/10.1016/j.lanepe.2021.100082>). Broadly speaking, if a symptomatic individual infects 4 susceptible persons, an asymptomatic individual would infect just 1. I am curious to know if the results are sensitive to this value, since children are likely to be asymptomatic. If feasible, the authors should consider a baseline scenario where the relative infectiousness of asymptomatic individuals matches the reported numbers.

- The authors state that the incubation period is a key parameter in their model as symptom-based surveillance is dependent on this period. However, it seems to be that the authors used a fixed period of 5 days in their model (Table S2) and was not subject to a sensitivity analysis. Given the importance of this parameter on model outcomes, might it be better to associate a distribution (say LogNormal?) and sample the incubation period for each infected individual? If this is not feasible, perhaps assessing the sensitivity might yield interesting results.

- Do the authors consider contact tracing in their model? For instance, when a symptomatic student is identified, their contacts should also be identified and isolated. Of course, this is not relevant in scenarios where a positive student triggers the closure of the entire school (and thus all students are isolated within their households).

- On line 90/91, it is stated that a symptomatic individual has a 50% chance of being tested. However, line 153/154 more precisely states the chance of getting tested is 95% for students and 50% for the general population. I think the language on 90/91 can be clarified to make this more explicit.

- The authors assume a 2-day period between sample collection and possible isolation. Is the student transmissible during these two days? I would imagine that a student would be sent home/isolated immediately at the onset of symptoms, but would return back to school if test shows negative.

- The discussion on the lack of vaccination for children is an important point (line 264). A natural question is then "what proportion of children should we identify to bring attack rates below a certain threshold", addressed by this study in JAMA: doi:10.1001/jamanetworkopen.2021.7097. Perhaps the authors can interpret their results within the context of the JAMA paper and include a citation.

- The relevant code was not provided at time of submission. It is important in the review process to be able to briefly look at the code to understand the model schematic as well as identify any bugs that may exist. I would recommend the authors to upload their codes to an online repository (also with explicit instructions on how to run the model, required dependencies/libraries, as well as the computational resources required).

Reviewer #4 (Remarks to the Author):

The manuscript by Dr. Liu and colleagues presents a computational model for the transmission dynamics of SARS-CoV-2, the pathogen responsible for the ongoing COVID-19 global pandemic. The model is specifically used to discuss different transmission control strategies for schools, with

application to Italy, where a reactive closure system based on symptom surveillance has been in place since September 2020. One of the main results from the model is that the strategy currently implemented in Italy is far from ideal, mostly because of the time delay between exposure and onset of infectiousness, on the one hand, and symptom onset and case isolation, on the other, which makes it difficult to promptly contain school outbreaks. The authors propose an alternative strategy that makes use of repeated screening of the school population using antigen tests. They show that a strategy like that could effectively prevent---or at least reduce---school-based COVID-19 outbreaks. The topic of the manuscript is clearly of paramount importance and likely of great interest to a wide audience of quantitative epidemiologists and decision-makers alike. From what I can gather, the analysis of the model is well done, and the results seem to be robust and quite convincing. The manuscript is also well written and easy to follow, despite the complexity of the underlying modeling framework. All that being said, I have a few technical comments that the authors may want to address while revising their manuscript.

Major comments

- The model is not exactly applied to the Italian case, rather it draws inspiration from it. By this, I mean that (i) no parameter calibration is performed and (ii) the model is applied to a synthetic population with realistic traits. I do not have objections regarding (i), as I understand that the main focus of this paper is not reproducing the temporal dynamics of the COVID-19 pandemic in Italy as it unfolded. Indeed, I do not have major objections regarding (ii) either, but here there are some things that I would like to understand better. Let's start with the abundance of the synthetic population: 500K people would correspond to a large Italian city (it would be the 7th largest, actually). Is this choice something demanded by computational feasibility, or could the numerical exercise potentially be scaled up to the ~60M people living in Italy? Are there any other issues preventing full country-scale implementation of the model? (Spatial connectivity and other possible heterogeneities come to mind, but there might be others). To be crystal clear: I am not saying that there is no value in the exercise if it does not go full country-scale; only, that I would like to understand the limitations of the approach (if any, computational or otherwise).

- Another thing that is not completely clear to me is to what extent the mixing pattern that has been assumed for community transmission can be deemed realistic. I understand that households and schools represent disconnected components within the overall interaction matrix, but what about the general community? I gather that this is described as a fully connected component, meaning---I think---that everyone is potentially in touch with anyone else through community mixing (perhaps following some age-specific rules). This is an assumption that is almost inevitably done in well-mixed, ODE models. However, I wonder whether the higher flexibility of agent-based modeling could allow for something different and, possibly, more realistic.

- Most of the model parameters are carefully set to match current knowledge on the transmission dynamics of SARS-CoV-2---when possible, with the Italian case study in mind. A parameter stands out, though, namely the probability of being tested if symptomatic (for both students and non-students). First, there is no clear definition of what "symptomatic" means. In many cases, this term is used with (not so slightly) different nuances, therefore its definition in the context of this paper should be clarified. Second, it would be interesting to understand where the proposed values of this parameter come from. I understand they are assumed and sensitivity analysis is performed, yet it would be great to have a glimpse of how the authors ended up proposing those figures. I am also asking this because of the remarkably different values used for students vs. non-students. Is school-based temperature control deemed so effective? Is it not implemented anywhere else, where community interactions would occur (e.g. places of business, transit stations)?

- I am a bit curious about how scenarios F50 and F25 are obtained. The generation of new infections can be seen as the outcome of a nonlinear, SIR-like process. Therefore, I wonder whether a unique configuration of the transmission parameters exists that can produce a given target scenario, or whether different combinations in the parameter space could lead to the same outcome, especially when two parameters are varied at the same time.

- In different panels of Fig.4 (as well as in the accompanying text), one gets the impression that higher levels of uncertainty are somehow associated with intermediate values of the basic

reproduction number. Are they? If so, why?

- Finally, I would like to understand whether the authors have considered comparing the outcomes of their proposed protocol for schools with an alternative scheme that uses the same testing effort and technology, yet applied to other components of the social mix---for instance the general community (one could think of testing workplaces at random, for instance). Given also the demographic differences between people mostly involved in school-based interactions compared to the general population, I think results might be non-trivial, in terms of both symptomatic cases and deaths averted. Of course, I understand that the focus of this manuscript is on schools and finding a safe protocol to let them open in pandemic times, yet I would be curious to see a comment by the authors on this point.

Minor comments

- l.38: identify -> identifying

- l.176: need to -> the; implement -> implementation of

- l.266: "it is still unclear... under 16 years" Well, that is kind of an understatement, considering that no trials have ever been conducted on children below the age of six---if I am not mistaken

- l.327: "Note that, to exclude... are considered" This disclaimer is repeated multiple times in the figure caption. If this is an important point, I would suggest expanding it in the Methods or the Supplements and removing it from most (if not all) captions

REVIEWER COMMENTS

Reviewer #1 (Remarks to the Author):

This is a nice analysis that uses an individual based model to estimate the impact of various school-oriented mitigation strategies on the overall transmission of SARS-CoV-2 in the community. They use a previously developed synthetic population to model transmission with explicit account of households and schools. The model structure is generated such that it resembles related data, but does not represent any particular population explicitly. They find that reactive class closure and interventions based on regular screening is more effective than quarantining based on syndromic testing. This paper addresses an important question; the approach is appropriate and the study is clearly communicated and well thought through. I have some comments that I recommend the authors address before this article is published.

We would like to thank the reviewer for taking the time to review our manuscript and for the constructive comments. We are glad that the reviewer believe that our approach is “appropriate and the study is clearly communicated and well thought through”.

Major comments:

1. Code availability:

It is becoming more usual that code is published alongside academic work for full transparency. I would appreciate access to the code to allow a full review of the work.

We are more than glad to provide the code to the reviewer. We have now included it as a zip file in the revised submission. Should the manuscript be published, we plan to post the code on a public repository (GitHub) as well.

2. Discussion of limitations.

There is currently very little discussion of how the model choices affect the results. Since this is an IBM, the outcome is very sensitive to the model specification and parameterisation. I would expect to see a detailed justification (including some in the main text) of these model choices and discussion of their impact on results. (I refer to some of these in more detail below)

We apologize for the lack of detail and discussion. We have heavily rewritten the Methods to detail the calibration of the model and justification of parameter choices as detailed in the responses to the following comments.

Epidemiological parameterisation:

The model relies heavily on it's specification and parameterisation. I think the authors could be clearer about the underlying epidemiological model in the main text. I also have some concerns about some of the epidemiological parameter ranges chosen. I would like to see more discussion of these choices and how they might affect the results.

We have now moved the description of the transmission model to the main text and included a new figure (Fig. 5, which is appended below for reviewer's convenience) showing a schematic representation of the model.

We have revised some of the parameter choices as suggested by the reviewer in her/his following comments. We have also added an extensive discussion on the effect of model parameters on the obtained results.

3. Given the recent changes in epidemiology due to new variants (particularly B.1.1.7 and B.1.617.2), I think it is now essential to discuss which variant this paper intends to study – although I imagine this was not the case when the work was carried out. A frustrating factor in modelling during a fast-moving pandemic.

Thank you for pointing this out. The work analyses the dynamics of SARS-CoV-2 in fall 2020, when historical lineages were circulating in Italy. We have now specified it in the introduction of the manuscript: “when the historical lineages were circulating in the fall of 2020” (line 71-72).

4. One key parameter, which is likely to affect results, is susceptibility of school aged children relative to adults. Broadly, this is still not clear in literature and therefore must be discussed as part of a work looking at interventions targeted at children. I would have expected this issue to be raised when presenting parameterisation in the main text, as well as discussion of how this might impact the results. The authors have based their estimate on the results of one study that uses data from very early in the pandemic. There have been a large number of studies using more recent data which may impact this choice and in turn the outcome of the study. Ideally, I would like to see more estimates applied in a sensitivity analysis, I recommend a discussion of this issue and how results might be affected.

We fully agree with the reviewer that the age-specific susceptibility to infection plays an important role in the analysis of age-targeted interventions. In the revised version of the manuscript, we use the susceptibility to infection by age as derived in a meta-analysis of the literature (Viner et al., JAMA Pediatrics, 2020) for the baseline analysis. In the supplementary material (Section 2.4), we show the results of a sensitivity analysis where all individuals are equally susceptible to infection (i.e., regardless of their age). Fig. S7 (also appended below for reviewer’s convenience) shows that the mitigation effect of the simulated class-closure strategy is consistent between the two scenarios.

Fig. S7. Sensitivity of the class-closure strategy based on syndromic surveillance to changes in susceptibility to infection by age.

The obtained results are mentioned in the main text as follows:

Line 155-157: “These results are also confirmed when we assume a shorter mean incubation period, homogeneous susceptibility to infection by age, and when symptomatic individuals are assumed to be twice or four times more infectious than asymptomatic individuals (Fig. S6-S8 in *SI Appendix*).”

5. Household SAR – it seems that this was derived from work unrelated to household attack rate. Household SAR has been estimated in a number of studies with results quite different from this (but again this depends on the variant the authors wish to model). Russel Viner has a systematic review of estimates. I recommend that this choice is reviewed, although I am not clear on the implications for the results.

We fully agree with the reviewer that a wide range of estimates of the household secondary attack rate (SAR) is available in the literature. The household SAR is highly dependent on the interventions adopted in a specific location (e.g., isolation of positive individuals in the place of residence as it is the case in Italy or in dedicated facilities as it is the case in China) as well as social, behavioral, and cultural characteristics of a population (e.g., number of persons per room in the house, which household member serves as a primary care giver inside the household, ventilation of the rooms). The estimates available in the literature are derived from contact tracing studies (most of them conducted in China and South-East Asia), in settings highly different from the Italian context. As such, we prefer to use an estimate (the only one, to the best of our knowledge) for the Italian population, which we have obtained in our previous study (Poletti et al., *JAMA Network Open*, 2021). We have added the following paragraph to justify our choice and acknowledge the literature on the topic:

Line 300-304: “Mixing patterns as well as social, behavioral, and cultural characteristics of the population (e.g., number of persons per room in the house, which household member serves as a primary care giver inside the household) have the potential to shape the household secondary attack rate. As such, we relied on an estimate available for Italy (about 50% ²¹) rather than estimates derived from contact tracing studies conducted in other countries ⁵²”

Minor comments:

1. The authors present the number of students infected before closing a class but it might be interesting to report the proportion of these in other classes.

As suggested, we have revised the figure, which now shows the proportion of students that are currently infectious in the other classes at the time a class closure is triggered. We have also replaced the figures in the Supplementary Material to adhere to this format.

2. It is interesting that more effective interventions (more frequent screening) results in more missed days, do the authors have a sense for when the interventions become effective enough that missed days reduce.

Thank you for this interesting comment that allowed us to clarify an important aspect of our analysis. The implemented strategy aims at reducing transmission rather than optimizing the number of missed school days and thus more frequent screening results in more frequent class closures. Once a class is closed, students of that class cannot go back to school for 14 days, regardless of their infectiousness. Moreover, even in the case that another student of a closed class was infected, she/he may become no longer infectious sooner than 14 days. If we were to allow earlier (than 14 days) return to schools for non-infectious students, a strategy based on more frequent screening could be both more effective in decreasing transmission and limiting the number of missed school days (as properly hinted by the reviewer’s comment). Such strategy could be implemented, for instance, through repeated PCR testing for quarantined and isolated students. However, such a strategy has never been considered as a possible option in Italy due to implementation difficulties. As such, we did not test this strategy in our analysis.

The following comment has been added to the main text to clarify this point:

Line 271-279: “Moreover, more frequent screening results in higher number of missed school days. The main contribution to this trend comes from the fixed duration of quarantine (14 days) of all students of a closed class, including those who tested negative. Also, students identified as positive during the screening had to remain in isolation for 14 days, regardless of whether they are still infectious or not. Repeated PCR testing of isolated and quarantined students would allow them to get back to school once they are confirmed as SARS-CoV-2 negative, decreasing unnecessary prolonged isolations and quarantines ⁴⁷. In this case, a strategy based on frequent screening may allow to both curtailing transmission and its associated burden in terms of missed school days. The number of missed school days could be further curtailed by requiring PCR confirmation of antigen positive samples as trigger for class-closures.”

3. The authors use a figure of 50% test rate for symptomatic cases in the community – is there a source for this?

We apologies for the lack of detail and we are indebted to the reviewer for noticing this omission. In the originally submitted version of the manuscript, we chose this parameter such that the case ascertainment ratio

for symptomatic individuals is 31.2% as reported in Marziano et al., PNAS, 2021 (<https://www.pnas.org/content/118/4/e2019617118>). Considering the changes in model parameterization asked by the reviewers, to obtain the same ascertainment ratio (i.e., 31.2%), the estimated test rate for symptomatic cases in the community is now set to 45%. We have added the following paragraph in the Methods section to clarify this point:

Line 379-383: “For the non-student population, symptomatic testing was not mandatory in Italy (except for specific workplaces such as health care and nursing home workers). As such, we considered that a symptomatic non-student individual has 45% probability of being tested. This parameter is set so that the case ascertainment ratio for any symptomatic individual in the overall population results to be 31%, matching the value reported in Marziano et al. ²³.”

4. Have the authors considered using within-school contact data from previous contact surveys to parameterise the model?

This is a very interesting point. The school structure and organization (e.g., phased school entries, limited group size in public areas of the school) has been deeply changed due to the COVID-19 pandemic. As of now, we are not aware of any study of the contact network of students in the Italian schools in the COVID-19 era. As such, we prefer to keep the model as simple as possible and considering only random mixing in the school and within each class, where the relative weight of these two components is derived from a pre-pandemic contact-survey study. We have now added the following paragraph to discuss this study limitation:

Line 288-292: “We consider only random mixing in the school and within each class, where the relative weight of these two components is derived from a pre-pandemic contact-survey study ⁵⁰. However, we acknowledge that the school structure and organization (e.g., phased school entries, limited group size in public areas of the school) has been deeply changed due to the COVID-19 pandemic. Unfortunately, as of October 2021, we are not aware of any study on the within-school and within-class contact network of students in the COVID-19 era in Italy.”

5. The model assumes homogenous mixing in the non-school, non-household population. This might impact the results as it may be expected that contacts cluster spatially which likely correlates with school populations. Could the authors comment on this? You might find this helpful for discussion of how schools interact through households <https://doi.org/10.1038/s41467-021-22213-0>.

This is another interesting point, and we fully agree with the reviewer that contacts may be spatially correlated with school populations (although it is unclear to what extent). Thank you for insightful reference; it has now been cited in the following paragraph of the Discussion where we comment on the limitations listed by the reviewer:

Line 288-304: “We consider only random mixing in the school and within each class, where the relative weight of these two components is derived from a pre-pandemic contact-survey study ⁵⁰. However, we acknowledge that the school structure and organization (e.g., phased school entries, limited group size in public areas of the school) has been deeply changed due to the COVID-19 pandemic. Unfortunately, as of October 2021, we are not aware of any study on the within-school and within-class contact network of students in the COVID-19 era in Italy. Moreover, the mixing patterns in the community have remarkably

changed as well. For example, mass gatherings (e.g., attending sport events, disco, cinema) were either banned for most of the duration of the pandemic or the capacity has been reduced, restrictions were imposed about the maximum number of people sitting at the same table in restaurants or allowed to enter commercial buildings at the same time, etc. As such, we kept the model as simple as possible, assuming homogenous mixing both in the school and community, although this represents a “first-order” approximation of the much more complex network of interconnections (e.g., between students attending different schools) ⁵¹. Nevertheless, the model has the flexibility to incorporate a more realistic representation of mixing patterns during the pandemic should new data become available for the focus location. Mixing patterns as well as social, behavioral, and cultural characteristics of the population (e.g., number of persons per room in the house, which household member serves as a primary care giver inside the household) have the potential to shape the household secondary attack rate. As such, we relied on an estimate available for Italy (about 50% ²¹) rather than estimates derived from contact tracing studies conducted in other countries ⁵²”

6. Line 95 - presumably the student isolates in the same way as scenario i. This could be clearer.

The reviewer is correct, and we apologize for the lack of clarity. This has been clarified in the entirely rewritten Methods section.

7. line 130: This seems more like a discussion point. Could be briefer here and more detailed in discussion. – there are a number of similar paragraphs in the results section.

As suggested, we have shortened paragraphs discussing our findings in the Results and extended the text related to these points in the Discussion.

8. I find the plots presenting the relative change in infections a bit un-intuitive – at first glance it looks like they are more effective for higher R values. Perhaps invert the axis?

Thank you for the suggestion. We have modified all plots accordingly.

9. I cannot find where you state the number of iterations of the model. I think this needs to be stated in each Figure caption to reflect the filtered samples.

We apologize for the lack of detail. Each scenario is based on 500 model runs. We have now added this information in the Methods section in the main text as well as to each figure caption.

10. Excluding simulations with under 5% infected to filter out spontaneous extinction seems sub-optimal (although I can see why you would do it). Could you instead simulate infections imported into the otherwise closed population (as randomly occurring infections) to protect against extinction altogether? This seems closer to reality.

We fully agree with the reviewer that continuous random importations would represent a situation closer to reality. As suggested, all the results of the revised version of the manuscript are based on a continuous importation of infected individuals. Specifically, at each simulated day, we sample the number of imported infectious individuals from a Poisson distribution of mean 1.34. This figure is based on the estimates reported by the national surveillance system: 0.027 imported cases per day per 10,000 individuals. We have also added a sensitivity analysis in the Supplementary Material (Sec. 2.2) showing to what extent obtained results are affected by this choice.

Reviewer #2 (Remarks to the Author):

An age stratified agent-based model was used to quantify the effects of testing through RT-PCR and school closures on community level transmission, hospitalization, and death. The results of the model suggest that screening students frequently with agent-based tests can be an effective measure to mitigate the burden of COVID-19. In general, I did not find too many major issues. As I read the manuscript, I felt there were parts of the methods and results that could be clarified for readers.

We would like to thank the reviewer for taking the time to review our manuscript and for the constructive comments that allowed us to improve the overall quality of our work.

Major

1. The sensitivity of the RT-PCR and antigen test was assumed to be static over the different stages of infection. Evidence suggests that the probability of detection is dependent on the stage of disease [Hellewell et al (2021) Estimating the effectiveness of routine asymptomatic PCR testing at different frequencies for the detection of SARS-CoV-2 infections, BMC Medicine ; Kucirka et al (2020) Variation in false-negative rate of reverse transcriptase polymerase chain reaction-based SARS-CoV-2 tests by time since exposure, Annals of Internal Medicine]. With the fixed sensitivity of test, the effectiveness of testing is possibly over-estimated.

We agree with the reviewer that the simple representation we used is an oversimplification of a more complicated process. As suggested, we have revised the model to include this feature. In particular, the sensitivity of the RT-PCR test now depends on the delay between infection and test according to the estimates provided in Kucirka et al (2020). Unfortunately, we were not able to find data about the sensitivity of the antigen test given the delay from the date of infection. For this reason, we decided to keep it fixed (at 69%) in the main text and perform a sensitivity analysis in the Supplementary Material (Sec. 4.2) where we assume that the antigen test follows the same temporal shape of sensitivity than the RT-PCR test, although with a lower absolute value.

For reviewer's convenience, we report here the figure showing the results of this sensitivity analysis (Fig. S15).

2. The specificity of the antigen test was estimated to be 99%, and then later assumed to be 100% for simplicity of the model. The issue here is that false positives will have implications on the specified interventions. Given the sheer volume of tests that would be conducted under the screening process, this assumption of 100% will have implications on the results. For example, conducting 100 tests on non-infected individuals that has a specificity of 99.9%, there is a 0.095 probability of at least one false positive.

We fully agree with the reviewer and would like to thank her/him for this comment. As suggested, we have revised the model to include false-positive results of the antigen test (1%). All the new results now account for this feature.

3. The incubation period was fixed at five days. Given some interventions are triggered by the presence of symptoms, accounting for heterogeneity in the incubation period could have a large impact on the level of uncertainty of the results. One could conduct a sensitivity analysis on the duration of the incubation period, integrate individual heterogeneity of the duration of the incubation period with a statistical distribution, or both (i.e. distributions with different means).

This is another interesting point. First, we have modified the model that now accounts for a distribution (gamma distribution of mean 6.2 days and standard deviation 4.3) of the incubation period. Second, we have conducted a sensitivity analysis where we have used an alternative duration of the incubation period (mean: 5.2 days). The obtained results are consistent with those obtained in the baseline analysis (see Fig. S6).

Minor

1. Line 86 – 87 is repetitive of prior lines 83-86.

Thank you – we amended the sentence to avoid repetitions.

2. Line 107: The mention of a 0.2-0.7 increase should be specified as an absolute increase to limit any confusion if it is relative.

Thank you – we have now specified that it is an absolute increase.

3. Instead of testing every student at one time one per week, the scenario of testing 1/7 of the student population every day was considered. This assumption implies that the school calendar was not considered. Given the flexibility of an agent-based model, this minor issue can easily be addressed. I expect that this small formality will not have substantial impact on the results.

We have revised the model according to this suggestion. All the simulations shown in the revised manuscript now consider 5 days of school and 2 days of weekend. As anticipated by the reviewer, this change had limited impact on the obtained results.

4. Line 404: The model is said to be simulated according to an SIR model, but the SARS CoV-2 is better defined by the SEIR model structure.

We have revised the model structure to account for a distribution of the incubation period. The new model has a clearly defined SEIR-like structure. We have revised the sentence accordingly as well as added a new figure (Fig. 5) to clarify the model structure.

5. Supplement pg 4: Infection is stratified into the infectious pre-symptomatic period, infectious symptomatic, infectious asymptomatic and recovered. However, there is no indication of the latent period (where the individual is infected but not infectious). It is unclear if this low level of infectivity is integrated into the parameter ϕ (infectiousness of individual j at time t), as I could not locate this function in Table S1. It is later specified to be chosen to reflect the generation time. This issue might be easily cleared up with an addition of a Figure to the SI.

The reviewer is correct. In the originally submitted manuscript, we accounted for an infectivity that depended on the time since infection (parameter ϕ). However, considering this and other reviewers' comments, we have modified the structure of the model, which now explicitly accounts for a latent period. This has now been clarified in the main text as well as in Fig. 5.

6. Figure 2. The aspects of the box plot are not described in the caption. I noticed it was specified in Figure 1 that it carries out, but this statement should be repeated for each caption.

Thank you for noticing this omission. The reviewer is correct – we used the same specification of the boxplot in all figures. In the revised manuscript, we have modified the definition to use the more canonical one (i.e., quantiles 0.025, 0.25, 0.5, 0.75, and 0.975). This information has been added to all captions.

7. The application of some of the probabilities should be clarified. For example, there could be confusion when sampling whether someone develops clinical symptoms or not and then later determining hospitalization and ICU (e.g. if my Bernoulli trial implies no symptoms, then does not seem reasonable that I go to the hospital). My impression is that these probabilities are used to assess hospitalization/deaths at the end of the simulation based on the cumulative number of infections. These calculations should be clarified in the SI.

We apologize for the lack of detail. The reviewer is correct. The number of individuals admitted to the hospital, developing critical disease, and deceased are estimated in post-processing based on the cumulative number of infections using Italy-specific estimates of: 1) infection hospitalization ratio, 2) infection critical illness ratio, and 3) infection fatality ratio (values reported in Tab. S2). We have now clarified this point in a separate paragraph in the Methods section:

Line 369-372: “To estimate COVID-19 burden, we leveraged the estimation of the infection hospitalization ratio, infection critical disease ratio, and infection fatality ratio obtained for Italy^{21, 54, 55} (Tab. S2 in *SI Appendix*). We applied them to the number of daily new infected individuals provided by the transmission model. Symptomatic individuals are instead calculated as described above.”

8. Simulations were initialized with a single case, but it was unclear whether simulations that quickly went extinct were discarded or not; and if not, the frequency in which early extinction occurred.

In light of another reviewer’s comment, in the revised version, infected individuals are imported every day according to a Poisson distribution of mean 1.34, based on the estimates reported by the national surveillance system (i.e., 0.027 imported cases per day per 10,000 individuals). As the result, in the revised manuscript, we do not have stochastic extinctions and all simulations are used in the analysis. We have also added a sensitivity analysis in the Supplementary Material (Sec. 2.2) showing to what extent the obtained results are affected by this choice.

9. In the text, it is unclear the number of ensembles the statistics are based on.

We apologize for the omission of this detail. Each scenario is based on 500 model runs. We have now added this information in the Methods section in the main text and the figure captions.

10. Figure 4B/E: I am particularly interested in the trend in the missed school days per student for the different scenarios as the effective reproductive number increases. For example, when $R=1.5$ the F50 scenario has fewer missed school days than the F25 but when $R=1.9$ the F50 scenario has more missed school days than the F25, but the F100 scenario is always substantially lower. More interestingly is the trend with antigen testing where the different testing frequencies have more missed school days under the different R values. I think it would be helpful to readers to briefly discuss why this trend is occurring.

Thank you for this interesting comment. There are several forces at play at the same time. On one hand, the larger R , the larger the number of infections and thus the larger the number of identified positive students triggering class closures. On the other hand, the larger R , the quicker the epidemic spreads and thus the shorter the duration of the epidemic, which also entails a shorter amount of time when class closures are triggered. Such processes occur under all scenarios of transmissibility in school (F25-F100) resulting in the trend described by the reviewer. However, the impact of the class-closure strategy heavily depends on the transmissibility among school-age individuals, thus altering the absolute number of missed school days per student.

The following comment has been added to the main text:

Line 193-200: “For scenarios F50 and F25, the impact of this strategy greatly depends on R , with reductions ranging from 20% and 70% (Fig. 4A). The number of missed school days per student greatly depends on R as well (Fig. 4B). The minimum number of missed school days per students is estimated for $R=1.3$ and F100 with 23.6 (95%CI: 18.1-29.3) days, while the largest value is estimated for $R=1.7$ and F50 with 60.3 (95%CI: 56.3-64.6) days. It is worth noting that there are several forces at play that determines the number of missed days. While larger values of R result in a larger number of infections (and thus the larger the number

of triggered class closures), it also leads to quicker epidemics spreads, thus reducing the time for class closures to be triggered.”

Regarding the second part of the comment by the reviewer. Once a class is closed, students of that class cannot go back to school for 14 days, regardless of their infectiousness. Moreover, even in the case that another student of a closed class was infected, she/he may become no longer infectious sooner than 14 days. If we were to allow earlier (than 14 days) return to schools for non-infectious students, a strategy based on more frequent screening could be both more effective in decreasing transmission and limiting the number of missed school days (as properly hinted by the reviewer’s comment). Such strategy could be implemented, for instance, through repeated PCR testing for quarantined and isolated students. However, such a strategy has never been considered as a possible option in Italy.

The following comment has been added to the main text to clarify this point:

Line 271-279: “Moreover, more frequent screening results in higher number of missed school days. The main contribution to this trend comes from the fixed duration of quarantine (14 days) of all students of a closed class, including those who tested negative. Also, students identified as positive during the screening had to remain in isolation for 14 days, regardless of whether they are still infectious or not. Repeated PCR testing of isolated and quarantined students would allow them to get back to school once they are confirmed as SARS-CoV-2 negative, decreasing unnecessary prolonged isolations and quarantines⁴⁷. In this case, a strategy based on frequent screening may allow to both curtailing transmission and its associated burden in terms of missed school days. The number of missed school days could be further curtailed by requiring PCR confirmation of antigen positive samples as trigger for class-closures.”

11. For the sensitivity analysis regarding immunity, it was assumed that 20% immunity was attained in all age classes. If immunity was shifted more so to the adults but still obtaining 20% immunity (i.e. if we think the children were sheltered from infection due to school closures), I expect transmission will become more prominent within the school

Thank you for this suggestion. We have added a new sensitivity analysis where we consider immunity to be age dependent. Specifically, we considered that the immunity in individuals aged 18 years or less is half of that for the rest of the population, by maintaining the total fraction of immune population at 20%. This new analysis is reported in Fig. S10 (which we include also here for reviewer’s convenience) and shows that the mitigation effect of the strategy is not very sensible to age-dependent heterogeneities in immunity.

Reviewer #3 (Remarks to the Author):

In this manuscript, the authors use an individual-based model to evaluate the impact of school closure on COVID-19 disease transmission. There have been a number of inconsistent studies regarding the efficacy of school closure, with inconsistencies arising from model assumptions and structures. In this manuscript, the described model is incredibly elaborate, and takes into account significant population heterogeneities that affect disease transmission and model outcomes. I commend them on their model structure and formulation, and have confidence in their results.

The authors find that school closures do reduce overall disease burden, but only to a mild degree (<15%), but can have a substantial impact on missed school days. In my interpretation, this is a very important point as missed school days may have significant socio-economical impact on the student as well as the general community, where this impact is possibly to be realized over the next few months or years.

Overall, I recommend this manuscript for publication but have a few minor concerns.

We would like to thank the reviewer for taking the time to assess our manuscript and her/his useful comments. We are honored the reviewer commend us on the model structure and formulation, and she/he has confidence in our results.

- The authors assume the infectiousness of asymptomatic individuals/students to be 100% relative to symptomatic infection. I am not sure if this assumption is justified (see: <https://doi.org/10.1016/j.lanepe.2021.100082>). Broadly speaking, if a symptomatic individual infects 4

susceptible persons, an asymptomatic individual would infect just 1. I am curious to know if the results are sensitive to this value, since children are likely to be asymptomatic. If feasible, the authors should consider a baseline scenario where the relative infectiousness of asymptomatic individuals matches the reported numbers.

We agree that the infectiousness of asymptomatic individuals with respect to symptomatic ones is a highly debated topic. To address the reviewer comment, we have added a sensitivity analysis where we assume the infectiousness of symptomatic individuals relative to asymptomatic ones to be twice and 4 times higher. The results of this sensitivity analysis are reported in Fig. S8, which we include here for reviewer’s convenience.

The obtained results are mentioned in the main text as follows:

Line 155-157: “These results are also confirmed when we assume a shorter mean incubation period, homogeneous susceptibility to infection by age, and when symptomatic individuals are assumed to be twice or four times more infectious than asymptomatic individuals (Fig. S6-S8 in *SI Appendix*).”

We have also extended the discussion on this topic, including the reference suggested by the reviewer:

Line 235-245: “Our results show that the reactive class-closure strategy implemented in the fall of 2020 has a limited potential in mitigating COVID-19 burden. This result is consistent when considering a wide set of sensitivity analyses on COVID-19 epidemiological characteristics (e.g., incubation period, age-specific susceptibility to infection, infectiousness of asymptomatic individuals relative to symptomatic ones, age-dependent heterogeneity in

population immunity, incubation), parameters regulating the implementation of the strategy (probability of testing symptomatic students and symptomatic non-student individuals, time intervals from symptom onset to sample collection or laboratory diagnosis), and model parameterization (daily imported infections to initialize the epidemic). Nonetheless, it is possible that the relative infectiousness of asymptomatic individuals could play an important role in the school transmission since children are more likely to be asymptomatic ²¹. However, our sensitivity analysis that considers asymptomatic individuals being two or four times less likely to transmit the virus ⁴¹, show a similar mitigation impact of the analyzed class-closure strategies.”

- The authors state that the incubation period is a key parameter parameter in their model as symptom-based surveillance is dependent on this period. However, it seems to be that the authors used a fixed period of 5 days in their model (Table S2) and was not subject to a sensitivity analysis. Given the importance of this parameter on model outcomes, might it be better to associate a distribution (say LogNormal?) and sample the incubation period for each infected individual? If this is not feasible, perhaps assessing the sensitivity might yield interesting results.

We fully agree with the reviewer about the importance of exploring the effect of the incubation period on the model outcome. First, we have modified the model that now accounts for a distribution (gamma distribution of mean 6.2 days and standard deviation 4.3) of the incubation period. Second, we have conducted a sensitivity analysis where we have used an alternative duration of the incubation period (mean: 5.2 days). The obtained results are consistent with those obtained in the baseline analysis (see Fig. S6).

- Do the authors consider contact tracing in their model? For instance, when a symptomatic student is identified, their contacts should also be identified and isolated. Of course, this is not relevant in scenarios where a positive student triggers the closure of the entire school (and thus all students are isolated within their households).

To resemble the epidemiological context analyzed in this manuscript (fall 2020 in Italy), a light form of contact tracing was considered in the model. Essentially, for each confirmed infection, her/his household contacts were traced and quarantined. This is implemented in the model as well. We apologize for the lack of clarity on this point. We have added the following paragraph to the Methods section to clarify how the model works:

Line 375-389: “The model explicitly simulates case isolation (in the place of residence), quarantine of household contacts (in the place of residence), and a reactive class-closure policy based on that implemented in Italy in the 2020-2021 school year. Case isolation and contacts quarantine are triggered by symptomatic individuals. Symptomatic students are tested with a 95% probability. Testing of symptomatic students was mandatory, but we considered that 5% of student population may not comply. For the non-student population, symptomatic testing was not mandatory in Italy (except for specific workplaces such as health care and nursing home workers). As such, we considered that a symptomatic non-student individual has 45% probability of being tested. This parameter is set so that the case ascertainment ratio for any symptomatic individual in the overall population results to be 31%, matching the value reported in Marziano et al. ²³. The sensitivity of the RT-PCR test depends on the delay between date of infection and test according to the estimates provided in Kucirka et al ⁵⁹. After sample collection and before the test result is obtained (2 days on average ⁶⁰), symptomatic individuals are precautionary quarantined in their place of residence and, if infectious, they can transmit to their household contacts only. Regardless of whether the positive individual is a student or not, they are isolated at home for

14 days starting with the date of laboratory confirmation. The household members of a positive individual are tested with RT-PCR and are quarantined at home for 2 weeks starting from the date of laboratory confirmation. Although mandatory, we considered a 95% compliance rate.”

- On line 90/91, it is stated that a symptomatic individual has a 50% chance of being tested. However, line 153/154 more precisely states the chance of getting tested is 95% for students and 50% for the general population. I think the language on 90/91 can be clarified to make this more explicit.

Thank you for pointing this out. We have amended the text as follows to avoid any confusion:

Line 87-88: “(i) active syndromic surveillance: a symptomatic (non-student) individual has a probability (45% in the baseline analysis) of being tested through PCR;”

- The authors assume a 2-day period between sample collection and possible isolation. Is the student transmissible during these two days? I would imagine that a student would be sent home/isolated immediately at the onset of symptoms, but would return back to school if test shows negative.

The reviewer is correct. The student is sent home and isolated. However, this student would still be able to transmit to her/his household members. This is now clarified in the text:

Line 384-387: “After sample collection and before the test result is obtained (2 days on average ⁶⁰), symptomatic individuals are precautionary quarantined in their place of residence and, if infectious, they can transmit to their household contacts only. Regardless of whether the positive individual is a student or not, they are isolated at home for 14 days starting with the date of laboratory confirmation.

- The discussion on the lack of vaccination for children is an important point (line 264). A natural question is then "what proportion of children should we identify to bring attack rates below a certain threshold", addressed by this study in JAMA: doi:10.1001/jamanetworkopen.2021.7097. Perhaps the authors can interpret their results within the context of the JAMA paper and include a citation.

Thank you for pointing us to this relevant study. We have revised the discussion to interpret our results in light of the suggested paper:

Line 258-261: “Our study supports the importance of student screening and testing for limiting silent transmission in schools and thus reduce the overall infection attack rate. As reported in other studies ⁴², the rapid identification of silent infections among children may be a valuable substitution of vaccination until a pediatric vaccine is available.”

- The relevant code was not provided at time of submission. It is important in the review process to be able to briefly look at the code to understand the model schematic as well as identify any bugs that may exist. I would recommend the authors to upload their codes to an online repository (also with explicit instructions on how to run the model, required dependencies/libraries, as well as the computational resources required).

We are more than glad to provide the code to the reviewer. We have now included it as a zip file in the revised submission. Should the manuscript be published, we plan to post the code on a public repository (GitHub) as well.

Reviewer #4 (Remarks to the Author):

The manuscript by Dr. Liu and colleagues presents a computational model for the transmission dynamics of SARS-CoV-2, the pathogen responsible for the ongoing COVID-19 global pandemic. The model is specifically used to discuss different transmission control strategies for schools, with application to Italy, where a reactive closure system based on symptom surveillance has been in place since September 2020. One of the main results from the model is that the strategy currently implemented in Italy is far from ideal, mostly because of the time delay between exposure and onset of infectiousness, on the one hand, and symptom onset and case isolation, on the other, which makes it difficult to promptly contain school outbreaks. The authors propose an alternative strategy that makes use of repeated screening of the school population using antigen tests. They show that a strategy like that could effectively prevent---or at least reduce---school-based COVID-19 outbreaks. The topic of the

manuscript is clearly of paramount importance and likely of great interest to a wide audience of quantitative epidemiologists and decision-makers alike. From what I can gather, the analysis of the model is well done, and the results seem to be robust and quite convincing. The manuscript is also well written and easy to follow, despite the complexity of the underlying modeling framework. All that being said, I have a few technical comments that the authors may want to address while revising their manuscript.

We would like to thank the reviewer for taking the time to review our manuscript, for the appreciation of our work, and for the insightful comments provided.

Major comments

- The model is not exactly applied to the Italian case, rather it draws inspiration from it. By this, I mean that (i) no parameter calibration is performed and (ii) the model is applied to a synthetic population with realistic traits. I do not have objections regarding (i), as I understand that the main focus of this paper is not reproducing the temporal dynamics of the COVID-19 pandemic in Italy as it unfolded. Indeed, I do not have major objections regarding (ii) either, but here there are some things that I would like to understand better. Let's start with the abundance of the synthetic population: 500K people would correspond to a large Italian city (it would be the 7th largest, actually). Is this choice something demanded by computational feasibility, or could the numerical exercise potentially be scaled up to the ~60M people living in Italy? Are there any other issues preventing full country-scale implementation of the model? (Spatial connectivity and other possible heterogeneities come to mind, but there might be others). To be crystal clear: I am not saying that there is no value in the exercise if it does not go full country-scale; only, that I would like to understand the limitations of the approach (if any, computational or otherwise).

All these considerations made by the reviewer are entirely correct. The main reason that prevents us from performing a country-scale analysis is the spatial connectivity of the Italian population during that phase of the pandemic. In the fall of 2020, several restrictions to the mobility between regions and between provinces were implemented with different level of intensity over different time windows; moreover, the behavior of the population in terms of travel patterns was far from pre-pandemic level. Overall, we strongly believe that we do not have enough data to provide a robust representation of the Italian mobility to develop a detailed national-scale Italian model. The second factor that advises us against using a country-level model is the computational time needed to run all the analyses. Indeed, the 12x factor needed to scale up the population is directly proportional to time required to run the simulations. Given the large spectrum of uncertainties surrounding the COVID-19 epidemiology, we favored the possibility to perform more comprehensive sensitivity analyses and provide outcomes in real time rather than performing a less refined analysis at the national level. These points are now discussed in Sec. Discussion as follows:

Line 306-314: “We built a synthetic population of about 0.5 million individuals (roughly the size of the sixth largest Italian city), rather than simulating the whole

Italian population. The main reason that prevents us from performing a country-scale analysis is the spatial connectivity of the Italian population during that phase of the pandemic. In the fall of 2020, several restrictions to the mobility between regions and between provinces were implemented with different level of intensity over different time windows; moreover, the behavior of the population in terms of travel patterns was far from pre-pandemic level. Overall, we believe that we do not have enough data to provide a robust representation of the Italian mobility to develop a detailed national-scale Italian model. The second factor that advises us against using a country-level model is the computational time needed to run all the analyses. Thus, we built a synthetic population well representative of a large Italian city.”

- Another thing that is not completely clear to me is to what extent the mixing pattern that has been assumed for community transmission can be deemed realistic. I understand that households and schools represent disconnected components within the overall interaction matrix, but what about the general community? I gather that this is described as a fully connected component, meaning---I think---that everyone is potentially in touch with anyone else through community mixing (perhaps following some age-specific rules). This is an assumption that is almost inevitably done in well-mixed, ODE models. However, I wonder whether the higher flexibility of agent-based modeling could allow for something different and, possibly, more realistic.

Ideally, we fully agree that the model has the flexibility to incorporate a more realistic representation of mixing patterns outside the household and school settings. However, we feel that the mixing patterns in the community have remarkably changed, and during the model design phase, we questioned the added value of incorporating high-resolution data on community mixing patterns collected before the COVID-19 pandemic. As such, we felt that assuming homogenous mixing was the best choice. We have added a discussion about this among the study limitations:

Line 288-304: “We consider only random mixing in the school and within each class, where the relative weight of these two components is derived from a pre-pandemic contact-survey study ⁵⁰. However, we acknowledge that the school structure and organization (e.g., phased school entries, limited group size in public areas of the school) has been deeply changed due to the COVID-19 pandemic. Unfortunately, as of October 2021, we are not aware of any study on the within-school and within-class contact network of students in the COVID-19 era in Italy. Moreover, the mixing patterns in the community have remarkably changed as well. For example, mass gatherings (e.g., attending sport events, disco, cinema) were either banned for most of the duration of the pandemic or the capacity has been reduced, restrictions were imposed about the maximum number of people sitting at the same table in restaurants or allowed to enter commercial buildings at the same time, etc. As such, we kept the model as simple as possible, assuming homogenous mixing both in the school and community, although this represents a “first-order” approximation of the much more complex network of interconnections (e.g., between students attending different schools) ⁵¹. Nevertheless, the model has the flexibility to incorporate a more realistic representation of mixing patterns during the pandemic should new data become available for the focus location. Mixing patterns as well as social, behavioral, and cultural characteristics of the population (e.g., number of persons per room in the house, which household member serves as a primary care giver inside the household) have the potential to shape the household secondary attack rate. As such, we relied on an estimate available for Italy (about 50% ²¹) rather than estimates derived from contact tracing studies conducted in other countries ⁵².”

- Most of the model parameters are carefully set to match current knowledge on the transmission dynamics of SARS-CoV-2---when possible, with the Italian case study in mind. A parameter stands out, though, namely the probability of being tested if symptomatic (for both students and non-students). First, there is no clear definition of what “symptomatic” means. In many cases, this term is used with (not so slightly) different nuances, therefore its definition in the context of this paper should be clarified. Second, it would be interesting to understand where the proposed values of this parameter come from. I understand they are assumed and sensitivity analysis is performed, yet it would be great to have a glimpse of how the authors ended up proposing those figures. I am also asking this because of the remarkably different values used for students vs. non-students. Is school-based temperature control deemed so effective? Is it not implemented anywhere else, where community interactions would occur (e.g. places of business, transit stations)?

We apologize for the lack of detail about this. Students are not allowed to enter the school building if they show any clinical sign or respiratory symptoms, and they are tested for SARS-CoV-2 infection. Specifically, these symptoms include dry cough, dyspnea, tachypnea, difficulty breathing, shortness of breath, sore throat, and chest pain or pressure; this definition has been added to the Methods section (**Line 364-366**). Instead of a 100% compliance to this regulation, we have arbitrarily considered a 95% compliance to this. Outside the school setting, except for specific workplaces (e.g., health care setting, nursing homes), there is no mandatory testing for workers and unoccupied individuals in case of symptom onset. We set the testing probability for symptomatic non-student individuals at 45%. This value was obtained to match the estimate case ascertainment ratio for symptomatic individuals in Italy, i.e., 31% (Marziano et al., PNAS, 2021 <https://www.pnas.org/content/118/4/e2019617118>). Note that in the originally submitted version of the manuscript we followed the same procedure and, to match the case ascertainment ratio for symptomatic individuals, we had to use a 50% testing probability for symptomatic non-student individuals. This small difference is likely due to the many changes in the model structure requested by the reviewers.

We have now clarified the testing procedures for these two segments of the population in the Methods section:

Line 377-383: “Symptomatic students are tested with a 95% probability. Testing of symptomatic students was mandatory, but we considered that 5% of student population may not comply. For the non-student population, symptomatic testing was not mandatory in Italy (except for specific workplaces such as health care and nursing home workers). As such, we considered that a symptomatic non-student individual has 45% probability of being tested. This parameter is set so that the case ascertainment ratio for any symptomatic individual in the overall population results to be 31%, matching the value reported in Marziano et al. ²³”

Regarding the sensitivity analysis proposed by the reviewer, we fully agree about its importance. This analysis is now reported in Fig. 3 where we have explored 80% testing probability for students as well as 20% and 70% testing probability for symptomatic non-student individuals. For reviewer’s convenience, we append Fig. 3 below.

Scenario F50

- I am a bit curious about how scenarios F50 and F25 are obtained. The generation of new infections can be seen as the outcome of a nonlinear, SIR-like process. Therefore, I wonder whether a unique configuration of the transmission parameters exists that can produce a given target scenario, or whether different combinations in the parameter space could lead to the same outcome, especially when two parameters are varied at the same time.

We apologize for the lack of clarity, and we would like to thank the reviewer for the opportunity to clarify this important aspect of our analysis. The calibration procedure works as follows. First, we calibrated the transmission rates in the household and community to obtain the household secondary attack rate available in the literature for Italy and the reproduction number observed in August-early September 2020 in Italy when the schools were closed for the summer break. Therefore, we have 2 free parameters and 2 conditions to meet. The estimation of the transmission rate in household could essentially be conducted regardless of the transmission in the community as the household secondary attack rate is not much influenced by the transmission outside the household. As such, in this first step of the calibration procedure, we have a unique configuration of model parameters matching the data. The second step of the calibration procedure is to calibrate the transmission rate in schools for the scenario F100. In this case, we are interested to match the reproduction number observed in Italy when schools reopened in the fall of 2020 while assuming that all the observed increase of transmissibility was linked to school transmission. Since in different Italian regions, we observed a different increase of transmissibility (ranging from 1.3 to 1.9), we have calibrated the transmission rate in school to reproduce 4 different values (1.3, 1.5, 1.7, and 1.9), while keeping the transmission rates in household and community to the values derived in the first step. For each value of R, we thus have only one configuration of parameters leading to the desired estimate. In the third and final step of the calibration procedure, we have recorded the number of infections generated in schools for scenario F100 and each value of R. Let us define this number of infections as f . In scenarios F50 and F25, we considered that only a certain fraction of the transmission increase observed in Italy after the reopening of schools was related to school transmission as other factors may have contributed as well (e.g., an increase in indoor leisure activities). In scenario F50, we consider that 50% (25% for the scenario F25) of f was related to school transmission. As such, we have kept the household transmission as estimated in step 1 and searched for the transmission in the community and in school to match the desired value of R and

number of infections linked to school transmission. Overall, also in this step there is a single configuration of model parameters able to match the aforementioned conditions.

We have added a specific section to the Methods (Sec. Model calibration and initialization) to clarify the procedure used to calibrate the mode.

- In different panels of Fig.4 (as well as in the accompanying text), one gets the impression that higher levels of uncertainty are somehow associated with intermediate values of the basic reproduction number. Are they? If so, why?

Thank you for this interesting comment. This pattern is no longer present in the revised simulations and disappeared once we used the revised model initialization: daily probability of importing cases vs. seeding the system with a given number of infectious individuals.

- Finally, I would like to understand whether the authors have considered comparing the outcomes of their proposed protocol for schools with an alternative scheme that uses the same testing effort and technology, yet applied to other components of the social mix---for instance the general community (one could think of testing workplaces at random, for instance). Given also the demographic differences between people mostly involved in school-based interactions compared to the general population, I think results might be non-trivial, in terms of both symptomatic cases and deaths averted. Of course, I understand that the focus of this manuscript is on schools and finding a safe protocol to let them open in pandemic times, yet I would be curious to see a comment by the authors on this point.

We would like to thank the reviewer for this interesting point of discussion. We agree that extending the screening to other contexts (e.g., workplaces) may represent a powerful mitigation/containment tool. We believe that, if this were to be considered by health officials, the analysis of the costs associated to this screening would be a key part of the analysis. As such, we defer it to a future study. We have added the following in Discussion to comment on this:

Line 327-328: “Moreover, our model is flexible to be extended to the screening in other contexts (e.g., workplaces) to provide insights on alternative mitigation/containment strategies outside of school context.”

Minor comments

- 1.38: identify -> identifying

Thank you – correction made.

- 1.176: need to -> the; implement -> implementation of

Thank you – correction made.

- 1.266: ”it is still unclear... under 16 years” Well, that is kind of an understatement, considering that no trials have ever been conducted on children below the age of six---if I am not mistaken

The sentence has been removed.

- 1.327: “Note that, to exclude... are considered” This disclaimer is repeated multiple times in the figure caption. If this is an important point, I would suggest expanding it in the Methods or the Supplements and removing it from most (if not all) captions

This is no longer the case in the revised manuscript as new infections are imported over time and thus, we no longer have stochastic extinction of simulated epidemics.

REVIEWERS' COMMENTS

Reviewer #1 (Remarks to the Author):

The work has been thoroughly revised. I am happy that the changes address my initial concerns sufficiently and would recommend this article for acceptance.

Reviewer #2 (Remarks to the Author):

1. I thank the authors for integrating my comment about accounting for heterogeneity in the incubation period by explicitly using a distribution. The distribution used is a Gamma with a mean of 6.2 and standard deviation of 4.3. However, in the main text of the reference publication (<https://doi.org/10.1038/s41467-021-21710-6>) a Weibull distribution is reported with a mean of 6.4 and the Gamma distribution reported in the SI for 114 clusters and 268 cases has a mean of 6.3. I expect the distribution determined from sensitivity analysis reported in Table S3 of the reference text? The authors should be quickly note why they chose to use the Gamma distribution from the sensitivity analysis in <https://doi.org/10.1038/s41467-021-21710-6> or describe how this distribution was constructed from the referenced publication.

2. For the sensitivity analysis of the duration of the incubation period, a reference is needed for the 5.2 day incubation period. If the average was only changed, and not the standard deviation, that should explicitly be indicated in the supplement.

Reviewer #3 (Remarks to the Author):

The authors have sufficiently address all my concerns. In addition, I reviewed comments by my peers and find that the authors have done a sufficient job in addressing their concerns as well (although it is up to them to ultimately decide). Although we are not at the tail-end of the pandemic with vaccines widely available (atleast in high-income countries), these results may not be directly applicable in the decision-making process. Nonetheless, it provides substantial value for future disease X outbreaks but also provides a proof of concept for the level of model complexity required to address meaningful questions.

All in all, I find that the authors have put in considerable efforts in the review of the manuscript and recommend it for publication.

Reviewer #4 (Remarks to the Author):

The authors have done a commendable job of answering the extensive set of comments and suggestions made by the referees during the first round of review, as well as of revising their manuscript accordingly. I particularly appreciated that changes were not simply "cosmetic"--- rather, the authors modified the structure of their model, updated some of the underlying assumptions, and basically re-run all of their analyses (plus some additional ones). I believe that both the presented results and the discussion revolving around them are stronger in the revised version of the paper than they already were in the original submission. This is all to say that I do not really have any additional comments at this point of the editorial process, and that I endorse the publication of this interesting work.

REVIEWER COMMENTS

Reviewer #1 (Remarks to the Author):

The work has been thoroughly revised. I am happy that the changes address my initial concerns sufficiently and would recommend this article for acceptance.

We are glad that the reviewer is pleased by our revision.

Reviewer #2 (Remarks to the Author):

1. I thank the authors for integrating my comment about accounting for heterogeneity in the incubation period by explicitly using a distribution. The distribution used is a Gamma with a mean of 6.2 and standard deviation of 4.3. However, in the main text of the reference publication (<https://doi.org/10.1038/s41467-021-21710-6>) a Weibull distribution is reported with a mean of 6.4 and the Gamma distribution reported in the SI for 114 clusters and 268 cases has a mean of 6.3. I expect the distribution determined from sensitivity analysis reported in Table S3 of the reference text? The authors should be quickly note why they chose to use the Gamma distribution from the sensitivity analysis in <https://doi.org/10.1038/s41467-021-21710-6> or describe how this distribution was constructed from the referenced publication.

We would like to thank the reviewer for their careful reading of our manuscript: we have corrected the 6.2 days into 6.3 days. To construct the distribution of the incubation period, we used a Gamma distribution with shape and scale parameters as reported in Hu et al., Nature Communications, 2021 (<https://doi.org/10.1038/s41467-021-21710-6>); this is now specified in the text.

2. For the sensitivity analysis of the duration of the incubation period, a reference is needed for the 5.2 day incubation period. If the average was only changed, and not the standard deviation, that should explicitly be indicated in the supplement.

We would like to thank the reviewer for noticing the missing reference. The estimate is taken from Zhang et al., Lancet Infectious Diseases, 2020 ([https://www.thelancet.com/journals/laninf/article/PIIS1473-3099\(20\)30230-9/fulltext](https://www.thelancet.com/journals/laninf/article/PIIS1473-3099(20)30230-9/fulltext)). This reference has been added to the supplement.

Reviewer #3 (Remarks to the Author):

The authors have sufficiently address all my concerns. In addition, I reviewed comments by my peers and find that the authors have done a sufficient job in addressing their concerns as well (although it is up to them to ultimately decide). Although we are not at the tail-end of the pandemic with vaccines widely available (atleast in high-income countries), these results may not be directly applicable in the decision-making process. Nonetheless, it provides substantial value for future disease X outbreaks but also provides a proof of concept for the level of model complexity required to address meaningful questions.

All in all, I find that the authors have put in considerable efforts in the review of the manuscript and recommend it for publication.

We would like to thank the reviewer for evaluating our manuscript and would like to thank them for the positive assessment.

Reviewer #4 (Remarks to the Author):

The authors have done a commendable job of answering the extensive set of comments and suggestions made by the referees during the first round of review, as well as of revising their manuscript accordingly. I particularly appreciated that changes were not simply "cosmetic"---rather, the authors modified the structure of their model, updated some of the underlying assumptions, and basically re-run all of their analyses (plus some additional ones). I believe that both the presented results and the discussion revolving around them are stronger in the revised version of the paper than they already were in the original submission. This is all to say that I do not really have any additional comments at this point of the editorial process, and that I endorse the publication of this interesting work.

We would like to thank the reviewer for praising our effort in revising the manuscript.